# PICKSTYLE: VIDEO-TO-VIDEO STYLE TRANSFER WITH CONTEXT-STYLE ADAPTERS

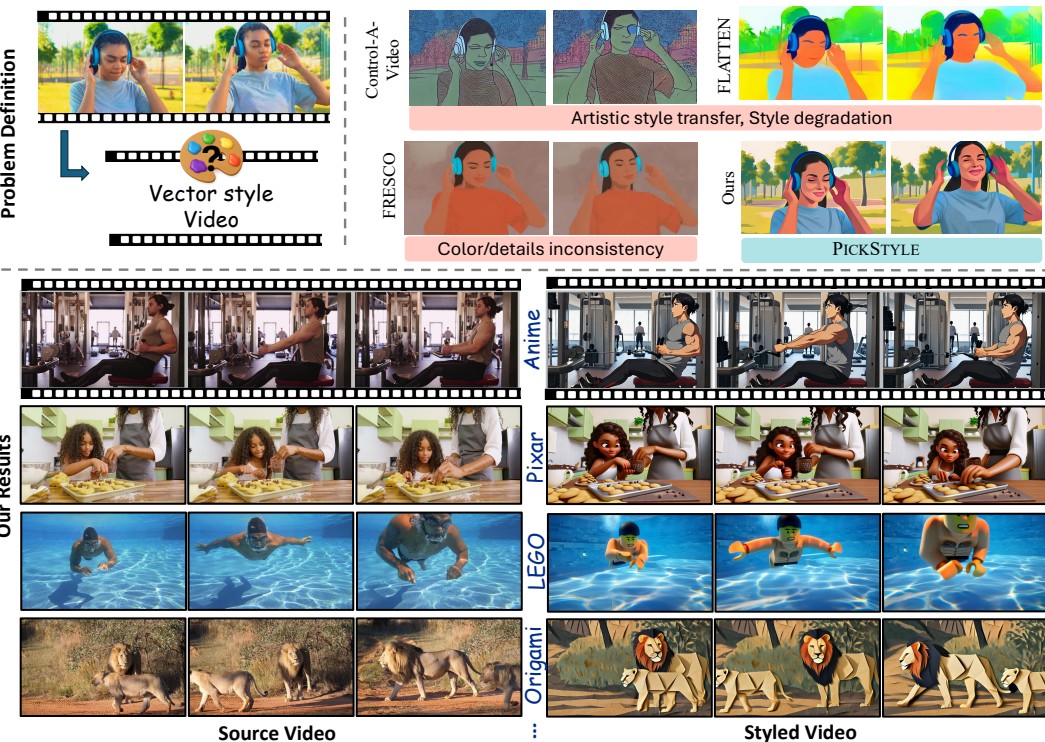

Figure 1: PICKSTYLE addresses video-to-video style transfer by preserving motion and context while translating videos into diverse styles. Unlike prior methods that treat the task as artistic style transfer (color–texture statistics while ignoring geometric properties of the target style) and that often suffer from style degradation, visual inconsistency and temporal flicker, PICKSTYLE produces coherent translations across nine styles.

## ABSTRACT

We address the task of video style transfer with diffusion models, where the goal is to preserve the context of an input video while rendering it in a target style specified by a text prompt. A major challenge is the lack of paired video data for supervision. We propose PICKSTYLE, a video-to-video style transfer framework that augments pretrained video diffusion backbones with style adapters and benefits from paired still image data with source–style correspondences for training. PICKSTYLE inserts low-rank adapters into the self-attention layers of conditioning modules, enabling efficient specialization for motion–style transfer while maintaining strong alignment between video content and style. To bridge the gap between static image supervision and dynamic video, we construct synthetic training clips from paired images by applying shared augmentations that simulate camera motion, ensuring temporal priors are preserved. In addition, we introduce Context–Style Classifier-Free Guidance (CS–CFG), a novel factorization of classifier-free guidance into independent text (style) and video (context) directions. CS–CFG ensures that context is preserved in generated video while the style is effectively transferred. Experiments across benchmarks show that our approach achieves temporally coherent, style-faithful, and content-preserving video translations, outperforming existing baselines both qualitatively and quantitatively.

# 1 INTRODUCTION

Recent advances in video diffusion models enable the generation of realistic, temporally coherent videos (Wan et al., 2025; Kong et al., 2024; HaCohen et al., 2024). Following these advances, a growing body of research explores ways to add controllability to text-to-video diffusion models, enabling finer-grained guidance over the generated content (He et al., 2025; Burgert et al., 2025; Jiang et al., 2025). While style transfer has advanced significantly for images, improvements in the video domain remain limited. This limitation is largely due to the scarcity of well-curated paired video datasets spanning diverse styles, in contrast to the abundance of such resources for images.

To mitigate data limitations, several methods (Yang et al., 2024; 2023) leverage image priors to apply style transfer on key frames and subsequently integrate them into videos, yet achieving coherent motion and appearance remains a persistent challenge. StyleMaster (Ye et al., 2025) synthesizes training data by leveraging the illusion property of VisualAnagrams (Geng et al., 2024), generating image pairs that share a common style while differing in content. Building on the still-moving paradigm, it subsequently trains a motion adapter on frozen video representations. Nevertheless, two key limitations remain. First, the synthetic pairs primarily capture artistic variations and are insufficient to model more complex styles, such as LEGO. Second, training a motion adapter on frozen videos presupposes a separation between spatial and temporal attention, whereas recent architectures (Wan et al., 2025; HaCohen et al., 2024; Kong et al., 2024) increasingly adopt spatiotemporal attention mechanisms, making such a decoupling more challenging.

To address these limitations, we exploit GPT-4o's (Achiam et al., 2023) strong style transfer capability to convert a Unity3D-rendered talk show into three distinct styles (anime, clay, and Pixar), thereby constructing a curated image dataset. We then augment this dataset with a subset of OmniConsistency (Song et al., 2025) to further increase stylistic diversity. To convert these image pairs into videos, we apply synthetic camera motions (e.g., zooming, sliding), creating sequences with simple movement and mitigating the risk of overfitting to static, motionless videos. Next, we keep the base model frozen and train a LoRA module on an auxiliary branch that conditions on RGB videos. To further strengthen this conditioning, we extend classifier-free guidance with context–style classifier-free guidance (CS-CFG), which jointly emphasizes the text prompt for style and the video for contextual information during denoising.

More concretely, our contributions are as follows: (1) We introduce a specialized and efficient adaptation of the VACE backbone by inserting LoRA modules into the spatiotemporal self-attention layers of the context branch. This enables effective motion-aware style transfer using *RGB conditioning*, a capability the frozen base model does not provide. (2) We propose CS-CFG, which factorizes the guidance into independent style (text) and content (RGB video) directions. By constructing a null context ($\mathcal{C}_{null}$) via spatiotemporal permutation, we ensure that content and temporal coherence are preserved while explicitly controlling style strength. (3) We mitigate the challenge of lacking paired stylized video data by introducing a solution that enables training on still images for moving-video stylization. we generate dynamic clips through synthetic camera motions, allowing the model to retain temporal priors and generalize from static image supervision to dynamic video content. (4) Through extensive experiments, we demonstrate that this combination yields strong geometric and stylistic transformations while maintaining temporal coherence and high fidelity to the conditioning video.

# 2 RELATED WORKS

We present core related work here and provide an extended discussion with details in the appendix K. Prior efforts in video style transfer span three main directions. Image-prior diffusion models (Zhang et al., 2023b; Yang et al., 2023; 2024; Cong et al., 2024) extend image diffusion to videos through temporal cues such as cross-frame attention, optical-flow propagation, or feature blending. Diffusion-based video editing methods (Wu et al., 2023; Qi et al., 2023; Ouyang et al., 2024) improve temporal consistency by aligning latent features or deformation fields across frames, but they primarily target text-guided editing rather than full video-to-video style transfer. Only a small set of works directly adapt video diffusion models for style transfer (Chen et al., 2023; Yue et al., 2025; Ye et al., 2025), typically relying on control signals or being suited for artistic style transfer. Large video diffusion backbones (Kong et al., 2024; HaCohen et al., 2024; Wan et al.,

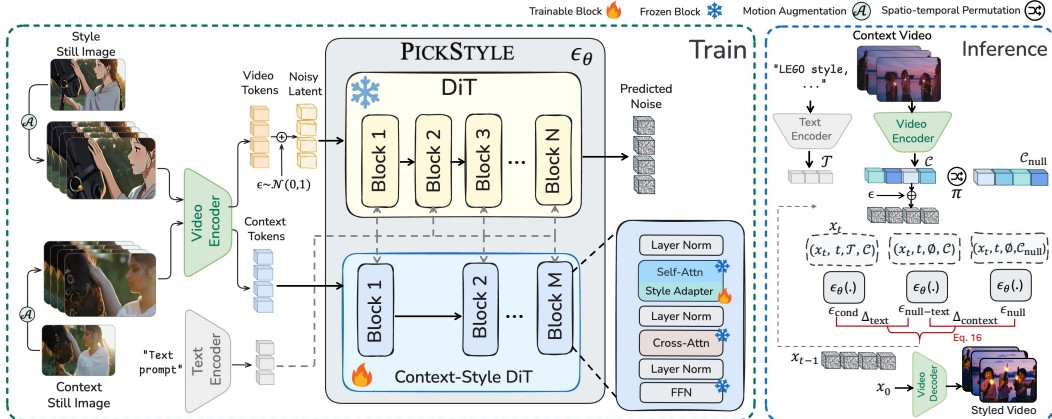

Figure 2: **Training and inference pipeline of PICKSTYLE.** In training (left), both the style image and the context image are transformed into video tokens and context tokens with synthetic camera motion using motion augmentation; video tokens are noised and denoised conditioned on context tokens by the DiT-based PICKSTYLE model with context-style adapters. In inference (right), a context video and a style description are encoded and iteratively denoised under text, context, and null conditions, where the proposed CS–CFG applies spatiotemporal permutation to the null context to generate the final styled video.

2025) demonstrate strong temporal coherence but are trained for text-to-video generation rather than video-conditioned style transfer.

## 3 PICKSTYLE

Our goal is to adapt text-to-video diffusion models for the task of video style transfer, where the content of an input video is preserved while its appearance is translated into a target style specified by a text prompt. A key challenge is the lack of paired video datasets for style transfer. To address this, we construct training data from pairs of images with different artistic or visual styles, which provide supervision for learning consistent appearance transformations.

### 3.1 PRELIMINARIES

**Conditional Diffusion Models** In conditional diffusion models, the forward process progressively corrupts a clean sample $x_0$ into a noisy latent $x_t$ through

$$q(x_t \mid x_{t-1}) = \mathcal{N}\big(x_t; \sqrt{\alpha_t}\, x_{t-1}, (1 - \alpha_t)\mathbf{I}\big), \tag{1}$$

until $x_T$ approximates Gaussian noise. The reverse process seeks to recover $x_0$ by denoising in a stepwise manner, modeled as

$$p_\theta(x_{t-1} \mid x_t, c), \tag{2}$$

where $c$ denotes the conditioning signal (e.g., class label, text, or image). This transition is parameterized by a neural denoiser $\epsilon_\theta(x_t, t, c)$ that predicts the injected noise at each step. Training minimizes the conditional objective

$$\mathbb{E}_{x_0, t, \epsilon}\left[\|\epsilon - \epsilon_\theta(x_t, t, c)\|^2\right], \tag{3}$$

ensuring that the learned reverse dynamics generate samples consistent with the condition $c$.

**Classifier Free Guidance.** Classifier-free guidance (CFG) is a widely used sampling technique that enhances the alignment of conditional diffusion models with a given condition $c$ without requiring an external classifier. Instead of relying solely on $\epsilon_\theta(x_t, t, c)$, the denoiser is jointly trained with and without conditions, yielding an unconditional branch $\epsilon_\theta(x_t, t, \varnothing)$. During inference, the two predictions are interpolated as

$$\hat{\epsilon}_\theta(x_t, t, c) = \epsilon_\theta(x_t, t, \varnothing) + \omega\big(\epsilon_\theta(x_t, t, c) - \epsilon_\theta(x_t, t, \varnothing)\big), \tag{4}$$

where $\omega > 1$ is the guidance scale. This formulation strengthens the influence of the condition by amplifying its contribution relative to the unconditional estimate, thereby producing samples that more faithfully follow $y$ while preserving sample diversity.

**VACE** Building on ACE (Han et al., 2024), VACE (Jiang et al., 2025) introduces multimodal input conditioning for text-to-video generation through the Video Condition Unit (VCU). Formally, VCU is defined as

$$V = (\mathcal{T}, \mathcal{F}, \mathcal{M}), \tag{5}$$

where $\mathcal{T}$ denotes the text prompt, $\mathcal{F} = \{u_1, u_2, \ldots, u_m\} \in \mathbb{R}^{C \times T \times H \times W}$ is the normalized video conditioning, and $\mathcal{M} = \{m_1, m_2, \ldots, m_n\} \in \{0,1\}^{T \times H \times W}$ is a binary mask, with 1 indicating tokens that can be modified and 0 indicating tokens that remain fixed. The model then computes reactive frames $\mathcal{F}_c = \mathcal{F} \odot \mathcal{M}$ and inactive frames $\mathcal{F}_k = \mathcal{F} \odot (1 - \mathcal{M})$, which are concatenated as $\mathcal{C} = [\mathcal{F}_c; \mathcal{F}_k]$ to form the final video conditioning input.

To inject the condition, VACE uses signals such as optical flow, depth maps, grayscale videos, scribbles, human 2D poses, and bounding boxes as $\mathcal{F}$ during training. Following ControlNet (Zhang et al., 2023a), it duplicates the pretrained text-to-video blocks into context blocks and trains them as a separate branch. These context blocks are fewer than the main blocks and skip certain layers, which makes the model more lightweight and improves convergence. The output of each context block is then added back to the corresponding DiT block in the main branch. While VACE incorporates diverse conditioning signals during training, RGB frames are always treated as inactive frames. As a result, the model can handle tasks such as inpainting and outpainting, but cannot encode RGB inputs as reactive frames, which limits its ability to perform tasks like style transfer. We introduce low-rank style adapter that converts RGB into a reactive conditioning path, enabling the backbone to propagate structural information from the input video and making style transfer feasible.

## 3.2 STYLE ADAPTER

Current video-to-video models usually rely on extra signals such as depth maps or optical flow derived from the original RGB video to guide the creation of new videos. This reliance creates a strong constraint because it makes it difficult to transfer styles like LEGO, where the low-level appearance changes significantly even though the overall scene and semantics remain the same. Fig. 5 shows an example in which VACE, using optical flow, fails to transfer the video into a 3D Chibi style because the spatial constraints prevent the model from adapting to the new structure. In addition, these modalities do not preserve the color information of each object.

Our approach adds a style adapter to the VACE context blocks, allowing the model to use RGB content as a conditioning signal and learn features that better support style transfer. Fig. 2 shows our pipeline. We adapt pretrained VACE model built on $N$ DiT blocks from Wan2.1, and adds $M$ context blocks ($M < N$) to encode the additional condition. We finetune only the self-attention layers of the context blocks. Cross-attention layers, which handle text conditioning, are left untouched because the model already demonstrates strong language understanding. Restricting adaptation to self-attention layers avoids disrupting the pretrained text-video alignment while still enabling the model to specialize in transferring motion and appearance across video domains.

Formally, the standard QKV projections in self-attention layers are defined as:

$$Q_i = W_Q Z_i, \quad K_i = W_K Z_i, \quad V_i = W_V Z_i, \quad i \in \{n, c\}, \tag{6}$$

where $Z_n, Z_c$ are input features for noise and context tokens, and $W_Q, W_K, W_V \in \mathbb{R}^{d \times d}$ are shared projection matrices used across all branches. We introduce LoRA transformations exclusively on the context blocks:

$$\Delta Q_c = B_Q A_Q Z_c, \quad \Delta K_c = B_K A_K Z_c, \quad \Delta V_c = B_V A_V Z_c, \tag{7}$$

where $A_Q, A_K, A_V \in \mathbb{R}^{r \times d}$ and $B_Q, B_K, B_V \in \mathbb{R}^{d \times r}$ are low-rank matrices with $r \ll d$.

The QKV for the context blocks is then updated as:

$$Q'_c = Q_c + \Delta Q_c, \quad K'_c = K_c + \Delta K_c, \quad V'_c = V_c + \Delta V_c, \tag{8}$$

while the noise branch remains unchanged:

$$Q'_n = Q_n, \quad K'_n = K_n, \quad V'_n = V_n. \tag{9}$$

### 3.3 TRAINING WITH IMAGE PAIRS

Due to advances in image models that enable style transfer Achiam et al. (2023) and the lack of video pairs, we use pairs of images to train our style adapter.. To enable the model to generalize from static image pairs to dynamic video content, we simulate motion during training. Specifically, we apply conventional data augmentations such as zooming in/out and sliding the crop window, which act as synthetic camera motions (see Fig. 8 for examples of the resulting training clips). For each image pair (source, style), we generate two corresponding video clips of length $T$ frames, where both clips undergo identical augmentation trajectories. This ensures the paired clips exhibit aligned synthetic motion while differing in style, allowing the model to learn temporal consistency during style transfer.

### 3.4 CONTEXT–STYLE CLASSIFIER-FREE GUIDANCE (CS–CFG)

Let $x_t$ denote the noised latent at diffusion step $t$, and let $\epsilon_\theta(x_t, t; \mathcal{T}, \mathcal{C})$ be the noise-prediction network conditioned on a text prompt $\mathcal{T}$ (style) and a video-conditioning tensor $\mathcal{C}$ (context). We construct a "null" version of the context by independently permuting its temporal and spatial axes. Concretely, if $\mathcal{C} \in \mathbb{R}^{t \times h \times w \times c}$ is the encoded context tensor in latent space, we draw independent uniform permutations $\pi_T \in S_T$, $\pi_H \in S_H$, $\pi_W \in S_W$, where $S_T$ (resp. $S_H$, $S_W$) denotes the symmetric group of all permutations of $\{1, \ldots, T\}$ (resp. $\{1, \ldots, H\}$, $\{1, \ldots, W\}$). The null context tensor is then defined as

$$\mathcal{C}_{\text{null}} = \pi_W \cdot \pi_H \cdot \pi_T \cdot \mathcal{C}, \tag{10}$$

with $(\pi_T \cdot \mathcal{C})_{t,h,w,c} = \mathcal{C}_{\pi_T(t),h,w,c}$ and analogously for $\pi_H$ and $\pi_W$. We then evaluate three forward passes:

$$\epsilon_{\text{cond}} = \epsilon_\theta(x_t, t; \mathcal{T}, \mathcal{C}), \tag{11}$$

$$\epsilon_{\text{null\_text}} = \epsilon_\theta(x_t, t; \varnothing, \mathcal{C}), \tag{12}$$

$$\epsilon_{\text{null}} = \epsilon_\theta(x_t, t; \varnothing, \mathcal{C}_{\text{null}}), \tag{13}$$

where $\varnothing$ denotes dropped text-conditioning (i.e., the classifier-free "null" token). CS–CFG factorizes the guidance into a *style (text) direction* and a *context (video) direction*:

$$\Delta_{\text{text}} = \epsilon_{\text{cond}} - \epsilon_{\text{null\_text}}, \tag{14}$$

$$\Delta_{\text{context}} = \epsilon_{\text{null\_text}} - \epsilon_{\text{null}}. \tag{15}$$

Given user-selected scales $t_{\text{guide}} \geq 0$ (style) and $c_{\text{guide}} \geq 0$ (context), the guided prediction is

$$\widehat{\epsilon} = \epsilon_{\text{null\_text}} + t_{\text{guide}} \Delta_{\text{text}} + c_{\text{guide}} \Delta_{\text{context}}. \tag{16}$$

### 3.5 NOISE INITIALIZATION STRATEGY

To enhance temporal coherence and preserve the context structure of the input video, we depart from the standard diffusion process that initializes sampling from pure Gaussian noise. Instead, we propose to initialize sampling from a *partially noised* version of the original video content $\mathcal{C}$. Given a total of $n$ denoising steps, we select a hyperparameter $k \in [1, n]$, and construct $x_{n-k}$ by applying the forward noising process to $\mathcal{C}$ up to step $n - k$:

$$x_{n-k} \sim q(x_{n-k} \mid x_0 = \mathcal{C}). \tag{17}$$

We then run the reverse process starting from $x_{N-k}$ down to $x_0$ using the DPM++ (Lu et al., 2025) sampler:

$$x_{t-1} = \text{DPM++}\big(x_t, \epsilon_\theta(x_t, t; \mathcal{T}, \mathcal{C})\big), \quad t = n - k, \ldots, 1, \tag{18}$$

where $\epsilon_\theta(x_t, t; \mathcal{T}, \mathcal{C})$ is the denoiser conditioned on the style prompt $\mathcal{T}$ and video content $\mathcal{C}$.

By initializing from $x_{n-k}$ rather than pure Gaussian noise, the model retains spatial and motion structure from the original video content $\mathcal{C}$, while still allowing sufficient stochasticity to adapt the style specified by $\mathcal{T}$. The hyperparameter $k$ controls the trade-off between style strength (larger $k$) and content/motion fidelity (smaller $k$).

Table 1: Quantitative comparisons on Content and Style Alignment across baselines and PICKSTYLE .

| Models | Content Alignment | | Style Alignment | | | | |
|---|---|---|---|---|---|---|---|
| | DreamSim ↓ | UMT ↑ | CLIP ↑ | CSD ↑ | R Precision ↑ | | |
| | | | | | Top@1 | Top@2 | Top@3 |
| Control-A-Video Chen et al. (2023) | 0.52 | 1.33 | **0.57** | 0.10 | 0.34 | 0.54 | 0.65 |
| Rerender Yang et al. (2023) | 0.41 | 2.47 | 0.55 | 0.13 | 0.27 | 0.39 | 0.54 |
| FLATTEN Cong et al. (2024) | **0.34** | 2.80 | 0.56 | 0.21 | 0.28 | 0.43 | 0.53 |
| FRESCO Yang et al. (2024) | 0.45 | 1.82 | 0.54 | 0.17 | 0.09 | 0.22 | 0.32 |
| PICKSTYLE | **0.34** | **3.33** | **0.57** | **0.37** | **0.75** | **0.85** | **0.91** |

## 3.6 SUMMARY OF TRAINING PROCEDURE

For clarity, we summarize the full pipeline here. During training, paired source–style images are converted into short synthetic video clips using shared motion augmentations, providing aligned appearance and motion supervision. The RGB clips are encoded as video-conditioning tokens and processed through the VACE context branch augmented with our LoRA-based style adapters. The diffusion model is trained to denoise the style clip conditioned on the source clip, learning coherent motion–style translation. During inference, the input video provides the context tokens, while a text prompt specifies the style. CS–CFG separates the text-driven and context-driven directions to preserve content while enforcing the target style.

## 4 EXPERIMENTS

**Implementation details.** We use the multi-node training framework of (Modal) with RDMA support to efficiently optimize the LoRA parameters. Our style adapter is trained on 32 H100 GPUs for 3000 steps with a learning rate of $5.6 \times 10^{-4}$ and rank $r = 128$ on the Wan2.1-VACE-14B variant with $M = 40$ DiT blocks and $N = 20$ context blocks and $d = 5120$. During inference, we apply $n = 20$ denoising steps with $t_{guide} = 5$ and $c_{guide} = 4$ in CS–CFG. To further improve results, we use TeaCache (Liu et al., 2025) to accelerate generation and APG (Sadat et al., 2024) to mitigate oversaturation. Additional details are provided in the appendix.

**Metrics.** We evaluate our method based on *Content Alignment*, *Style Alignment*, and *Video Quality*. For content alignment, we compute frame-level similarity using the DreamSim (Fu et al., 2023) distance between corresponding frames in the original and generated videos, and report the final score by averaging across all frames. We further evaluate how well the generated video matches its high-level text description using UMTScore (Liu et al., 2023). For style alignment, we calculate the CLIP score (Hessel et al., 2021) between each generated frame and a textual style prompt, then average over frames to obtain the final score. We also compute the CSD score (Somepalli et al., 2024) by first averaging the similarity between each generated frame and the target style exemplars, and then averaging across frames to produce the overall style alignment score. We further evaluate top-$k$ R Precision using Gemini (Team et al., 2023) by classifying the middle frame of each generated video against all candidate style prompts. For each frame, Gemini returns the top-$k$ most likely styles in order, and we compute top-$k$ precision for each frame, and averaging across frames to produce the final precision score. For Video quality, we use Motion smoothness, dynamic quality, and visual quality from VBench (Huang et al., 2024) benchmark. Motion smoothness leverages the motion priors in the AMT (Li et al., 2023) model to leverage the smoothness of generated videos. Dynamic quality uses RAFT (Teed & Deng, 2020) to estimate degree of dynamics, and Visual quality uses MUSIQ (Ke et al., 2021) on each frame to assess distortions such as over-exposure, noise, or blur.

**Dataset.** Our training dataset consists of paired images across multiple styles. We begin by extracting 250 diverse frames from an animated 3D talk show rendered in Unity3D, which serve as our source images. Using GPT-4o, we transform each frame into three distinct styles: Anime, Pixar, and Claymation. To ensure consistency in content between the generated samples and the originals, we manually refine the prompts for each case. This process yields a *carefully curated* dataset of 750 stylized samples, containing both the original reference frames and their three stylistic variants. To further enhance the diversity of training data, we incorporate six styles from OmniConsistency's dataset (Song et al., 2025): 3D Chibi, Vector, LEGO, Rick & Morty, Origami, and Macaron, and we further augment our Claymation style using their samples.

Table 2: Quantitative comparisons on Video Quality metrics across baselines and PICKSTYLE

| Models | Video Quality | | | Overall |
|---|---|---|---|---|
| | MotionSmooth ↑ | DynamicQuality ↑ | VisualQuality ↑ | |
| Control-A-Video Chen et al. (2023) | 0.976 | 0.602 | 0.683 | 0.754 |
| Rerender Yang et al. (2023) | 0.990 | 0.667 | 0.567 | 0.741 |
| FLATTEN Cong et al. (2024) | 0.977 | 0.780 | 0.592 | 0.783 |
| FRESCO Yang et al. (2024) | **0.993** | 0.632 | 0.623 | 0.716 |
| PICKSTYLE | 0.982 | **0.797** | **0.688** | **0.822** |

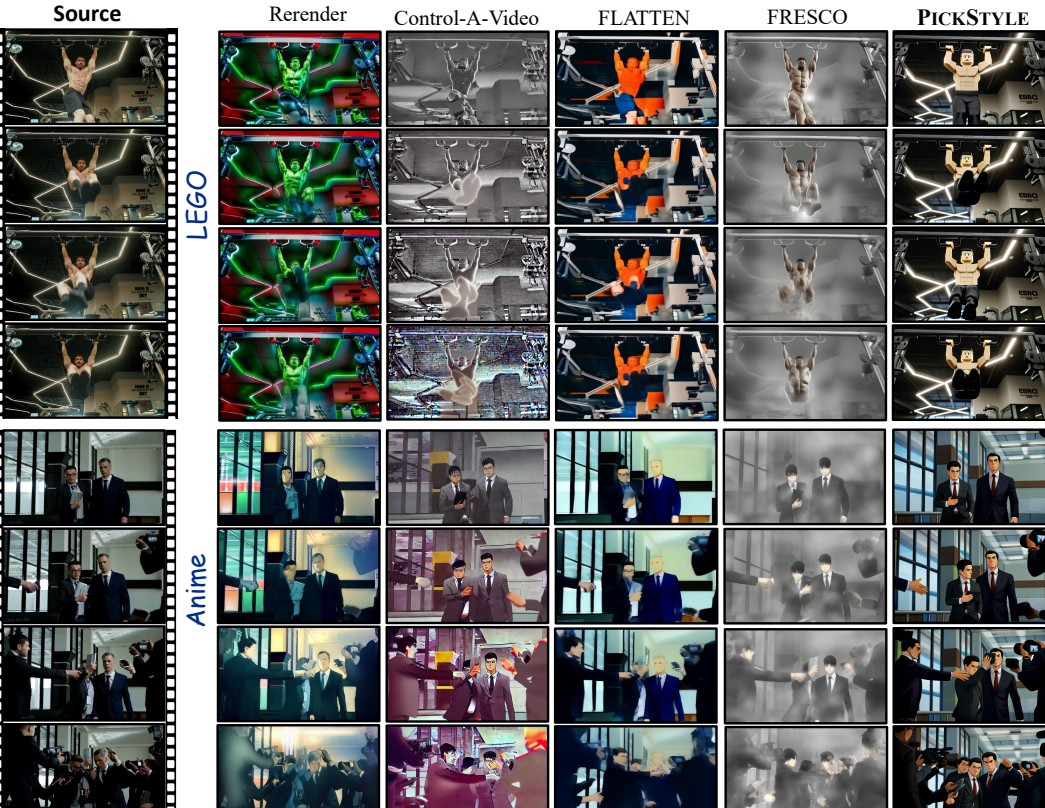

Figure 3: Qualitative comparison of PICKSTYLE , Control-a-Video, Rerender, FRESCO, and FLATTEN in LEGO and anime styles.

## 4.1 COMPARISONS WITH OTHER METHODS

We evaluate PICKSTYLE against the main classes of existing methods that can operate on *video-to-video* style transfer. These include (i) image-prior diffusion approaches (Yang et al., 2023; Cong et al., 2024; Yang et al., 2024), which dominate current video stylization practice and represent the strongest publicly available solutions; and (ii) *video-diffusion-based baseline* (Chen et al., 2023; Jiang et al., 2025).

**Quantitative comparison.** Table 1 compares PICKSTYLE with prior approaches on both content and style alignment metrics. For content alignment, PICKSTYLE achieves the lowest DreamSim score (0.34) *and* the highest UMTScore (3.33), indicating stronger frame-level consistency and better alignment with high-level content descriptions than the baselines. On style alignment, PICKSTYLE reaches the highest CSD score (0.37). While CLIP score remains tied with Control-A-Video (0.57), PICKSTYLE achieves substantially higher R Precision across all top-$k$ levels, demonstrating more accurate alignment with the target styles.

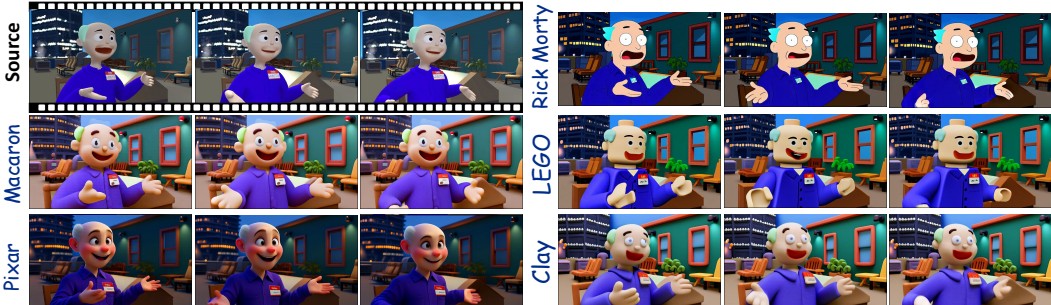

Figure 4: Qualitative evaluation of PICKSTYLE on a non-photorealistic example rendered in Unity3D.

Table 2 demonstrates that PICKSTYLE achieves a clear margin over existing approaches in both dynamic quality *and* visual quality, the two metrics most reflective of temporal coherence and perceptual fidelity. MotionSmooth remains nearly perfect for all methods, since they are derived from video-to-video models that inherently preserve motion trajectories, and the small numerical differences are therefore negligible. When aggregated, PICKSTYLE obtains the highest overall score, highlighting its effectiveness in generating temporally consistent and perceptually compelling video outputs compared to prior work.

**Qualitative comparison.** Fig. 3 presents a qualitative comparison of PICKSTYLE with Rerender, Control-a-Video, FLATTEN, and FRESCO on LEGO and Anime styles. The competing methods, which rely on depth maps or HED edges (Xie & Tu, 2015) as inputs, lack access to color information, often producing mismatched hues and noticeable color artifacts in their generated videos. In addition, Rerender and FRESCO, being image-based models, exhibit poor temporal consistency and suffer from frame-to-frame flickering. Finally, while the geometry constraints in these baselines sometimes succeed in forming LEGO-like structures in local regions such as the head, they frequently fail to propagate these stylistic details across the entire body. In contrast, PICKSTYLE consistently delivers faithful color reproduction, stable temporal coherence, and coherent geometry throughout the video. Additional qualitative comparison results across styles are provided in the Appendix and supplemental video.

Fig. 4 shows qualitative results on Unity3D animations that we collected and used to train Anime, Pixar, and Clay styles. Although this dataset differs from the photorealistic data used to train other styles, PICKSTYLE is still able to transfer styles such as LEGO, Rick & Morty, and Macaron from OmniConsistency, which were originally trained on photorealistic counterparts. This demonstrates that PICKSTYLE generalizes effectively across domains, handling both photorealistic and non-photorealistic inputs. Moreover, it highlights a practical application for animated content: instead of depending on high-quality outputs from 3D engines, one can rely on simple Unity3D renderings and leverage style transfer to achieve visually compelling results.

In Fig. 5, we further compare PICKSTYLE with VACE on 3D Chibi style generation. Here, optical flows extracted using RAFT (Teed & Deng, 2020) serve as the input condition for VACE. Because these flows do not contain color information, VACE cannot preserve the lost appearance details in its outputs. In addition, since VACE was not originally designed for style transfer and is highly sensitive to the input geometries, it struggles to capture the intended stylistic patterns and fails to achieve reliable style transfer. More extensive comparisons with alternative input modalities supported by VACE are provided in the Appendix.

## 4.2 ABLATION STUDIES

**Effect of motion augmentation.** Fig. 6 shows the effect of motion augmentation on videos generated by PICKSTYLE in anime and Pixar styles. For the anime samples both the video description and the style prompt are provided, while for the Pixar samples only the style prompt is given. When the video description is included the generated results achieve both good motion quality and faithful style transfer. Without motion augmentation however small background motions such as people walking on a treadmill are often missed, as the model pays less attention to fine motion details. The gap becomes larger when the video description is not provided. In the Pixar example the model without motion augmentation cannot fully preserve actions such as the jump at the end of the video

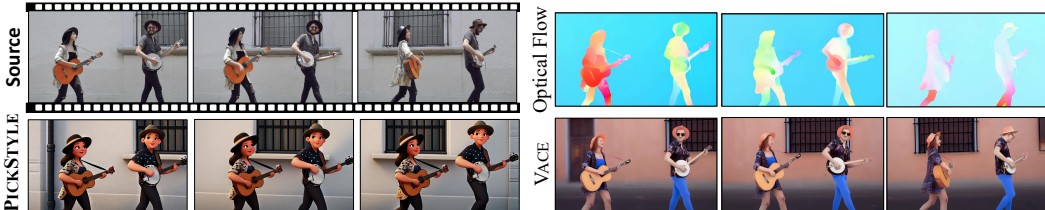

Figure 5: Comparison between PICKSTYLE and the VACE baseline in 3D Chibi style. VACE fails to capture the target style.

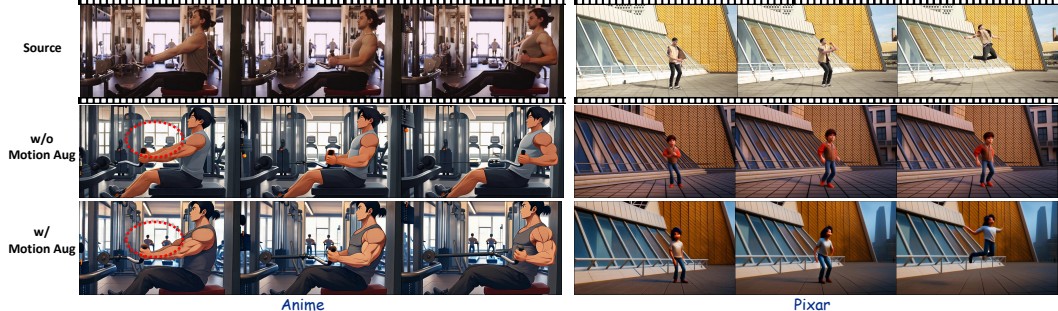

Figure 6: Effect of motion augmentation of generated video of PICKSTYLE .

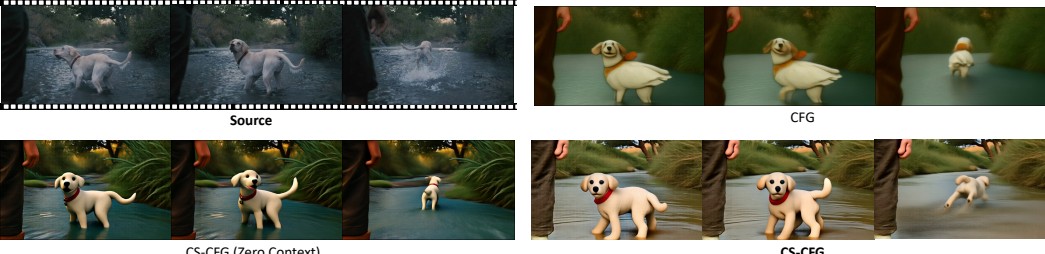

Figure 7: Effect of CS-CFG on the style transferring, evaluated on Clay style.

and focuses mostly on style transfer. With motion augmentation the model better captures both large scale and subtle motions even when detailed descriptions are not available.

**Effect of CS–CFG.** Fig. 7 highlights the effectiveness of CS-CFG in improving style transfer. With CFG, only the style guidance in text prompt influences the output, so while the video carries the intended clay style, it lacks fidelity to the original content. In this case, the model confuses the dog with a swan due to its generative prior and produces a hybrid appearance that diminishes contextual accuracy. An alternative design replaces the null video context in CS-CFG with zero pixels, which yields partial improvement over CFG but results in oversaturation and incomplete preservation of the clay style, as seen for instance in the person's hand where fine details are lost. In contrast, CS-CFG leverages spatiotemporal permutation to better capture contextual cues, leading to sharper details, faithful clay-style transfer, and stronger adherence to the intended content.

## 5 CONCLUSION

We introduced PICKSTYLE , a video-to-video style transfer framework built on VACE with context–style adapters and a novel CS–CFG mechanism. Despite being trained on a relatively limited dataset, PICKSTYLE effectively preserves motion and context while rendering diverse target styles. By leveraging synthetic motion-augmented training pairs and a noise initialization strategy, it achieves superior style fidelity, temporal stability, and perceptual quality compared to existing methods. Beyond quantitative improvements, PICKSTYLE consistently produces coherent color reproduction and faithful geometry across diverse styles while avoiding the temporal flicker and blending artifacts common in image-based approaches. These results highlight that even with constrained supervision, PICKSTYLE can deliver high-quality style transfer and establish a strong baseline for future research in controllable video stylization.

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

APPENDIX

**Supplemental Video.** The supplemental video provides qualitative demonstrations that illustrate the effectiveness of our approach across various styles and scenarios. We strongly encourage readers to view the supplemental video for a more comprehensive understanding of the results.

## A  AUGMENTATION DETAILS

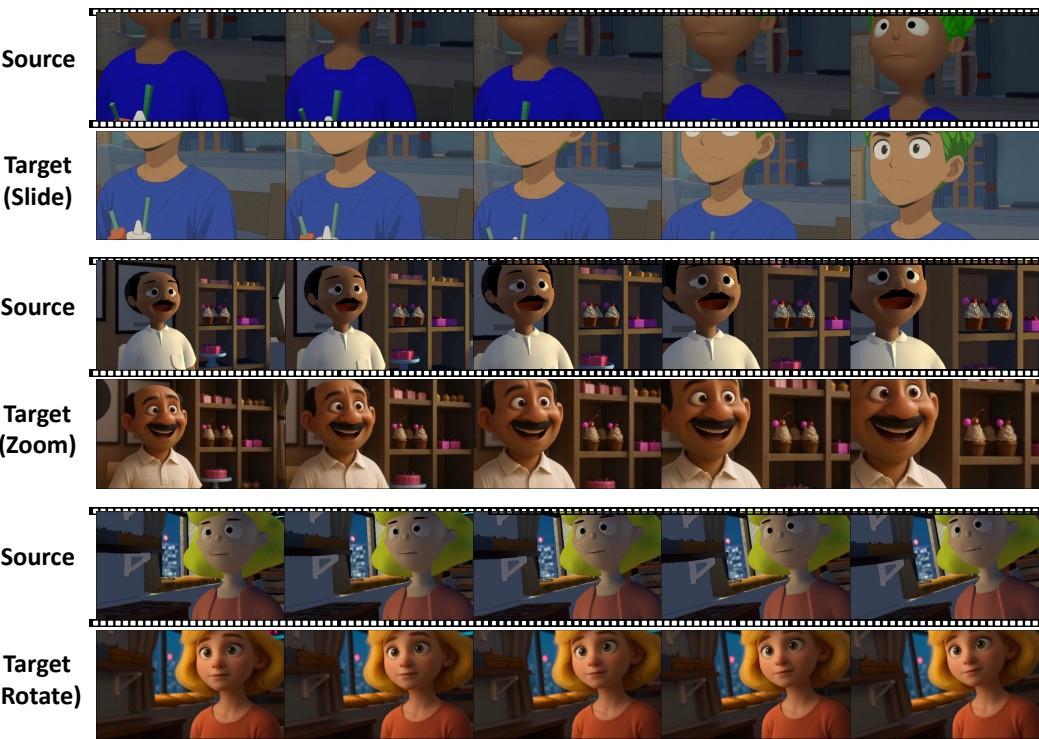

Figure 8: Examples of source and target samples throughout Style Adapter training.

To train the style adapter using image pairs, we synthetically add camera motion and create pairs of videos with synthetic camera movements. Although simple, these motions help the style adapter to preserve the motion prior and transfer the style from the conditioning video while preserving the motion. We use zoom in, zoom out, sliding in four directions, and rotation as synthetic motion augmentation. Fig. 8 shows source and target samples in training made of sliding, zooming, and rotating the input and target images. Below, we explain in detail how the augmentation is applied throughout training.

**Zoom.** let $X \in \mathbb{R}^{C \times T \times H \times W}$ be the input video. The zoom factor is sampled as $z \sim \mathcal{U}(1.2, 2.0)$, and the zoom mode is chosen at random as $m \sim \{\text{in, out}\}$. For each frame index $t \in \{0, \ldots, T-1\}$, the per-frame scale is

$$s_t^{(m)} = \begin{cases} z^{\frac{t}{T-1}}, & m = \text{in} \quad \text{(zoom-in)}, \\ z^{1-\frac{t}{T-1}}, & m = \text{out} \quad \text{(zoom-out)}. \end{cases}$$

The corresponding crop size is

$$h_t^{(m)} = \frac{H}{s_t^{(m)}}, \qquad w_t^{(m)} = \frac{W}{s_t^{(m)}},$$

and the (centered) crop offsets are

$$\Delta y_t^{(m)} = \frac{H - h_t^{(m)}}{2}, \qquad \Delta x_t^{(m)} = \frac{W - w_t^{(m)}}{2}.$$

Let $X_t$ denote the $t$-th frame, i.e. $X_t(c, y, x) = X(c, t, y, x)$. We first take a central crop

$$X_t^{\text{crop},(m)}(c, u, v) = X_t\big(c,\ u + \Delta y_t^{(m)},\ v + \Delta x_t^{(m)}\big), \quad u \in [0, h_t^{(m)}),\ v \in [0, w_t^{(m)}),$$

and then resize it back to $(H, W)$ using bilinear interpolation $\mathcal{R}_{H,W}(\cdot)$. The output video $Y^{(m)} \in \mathbb{R}^{C \times T \times H \times W}$ is

$$Y^{(m)}(c, t, y, x) = \mathcal{R}_{H,W}\Big(X_t^{\text{crop},(m)}(c, \cdot, \cdot)\Big)(y, x), \quad y \in [0, H),\ x \in [0, W).$$

**Slide.** we reuse the zoomed crop but translate it over time. Let $X \in \mathbb{R}^{C \times T \times H \times W}$ be the input video. We apply a fixed zoom factor $z_{\text{slide}} > 1$ (we use $z_{\text{slide}} = 1.2$ in our experiments). The maximum shift is sampled at random as

$$M \sim \mathcal{U}\{100,\ 200\},$$

and the slide direction is chosen at random as

$$d \sim \{\text{right, left, up, down}\}.$$

For each frame index $t \in \{0, \dots, T-1\}$, we define a time-dependent shift

$$\tau_t = \frac{t}{T-1}, \qquad \delta_t = M\,\tau_t^2,$$

and a zoomed crop size

$$h = \frac{H}{z_{\text{slide}}}, \qquad w = \frac{W}{z_{\text{slide}}}.$$

We define the base (non-sliding) offsets

$$\Delta y^{\text{center}} = \frac{H-h}{2}, \qquad \Delta x^{\text{center}} = \frac{W-w}{2}, \qquad \Delta x^{\text{right}} = W - w.$$

The time-varying crop offsets $(\Delta y_t^{(d)}, \Delta x_t^{(d)})$ for each direction $d$ are

$$(\Delta y_t^{(d)}, \Delta x_t^{(d)}) = \begin{cases} \big(\Delta y^{\text{center}},\ \delta_t\big), & d = \text{right}, \\ \big(\Delta y^{\text{center}},\ \Delta x^{\text{center}} + (W - w - \Delta x^{\text{center}}) - \delta_t\big), & d = \text{left}, \\ \big((H - h) - \delta_t,\ \Delta x^{\text{right}}\big), & d = \text{up}, \\ \big(\delta_t,\ \Delta x^{\text{right}}\big), & d = \text{down}. \end{cases}$$

Let $X_t$ denote the $t$-th frame, $X_t(c, y, x) = X(c, t, y, x)$. We extract a sliding crop

$$X_t^{\text{slide},(d)}(c, u, v) = X_t\big(c,\ u + \Delta y_t^{(d)},\ v + \Delta x_t^{(d)}\big), \quad u \in [0, h),\ v \in [0, w),$$

and resize it back to $(H, W)$ using bilinear interpolation $\mathcal{R}_{H,W}(\cdot)$. The final output video $Y^{(d)} \in \mathbb{R}^{C \times T \times H \times W}$ is

$$Y^{(d)}(c, t, y, x) = \mathcal{R}_{H,W}\Big(X_t^{\text{slide},(d)}(c, \cdot, \cdot)\Big)(y, x), \quad y \in [0, H),\ x \in [0, W).$$

**Rotation.** we apply a smooth temporal rotation to the video frames. Let $X \in \mathbb{R}^{C \times T \times H \times W}$ be the input video. The maximum rotation angle is sampled as $\theta_{\max} \sim \mathcal{U}(-20°, 20°)$, since larger magnitudes expose the image borders and introduce empty background regions. We additionally apply a fixed zoom factor $z_{\text{rot}} = 1.5$ to crop out black edges after rotation.

For each frame index $t \in \{0, \dots, T-1\}$, the rotation angle evolves linearly:

$$\theta_t = \theta_{\max} \frac{t}{T-1}.$$

Let $X_t(c, y, x) = X(c, t, y, x)$ denote the $t$-th frame, and let $\mathcal{R}_\theta(\cdot)$ be a rotation operator (bilinear sampling, zero-filled outside bounds). The rotated frame is

$$\tilde{X}_t(c, y, x) = \mathcal{R}_{\theta_t}(X_t)(c, y, x).$$

To remove rotation-induced background, we crop a centered region of size

$$h = \frac{H}{z_{\text{rot}}}, \qquad w = \frac{W}{z_{\text{rot}}}, \qquad \Delta y = \frac{H - h}{2}, \qquad \Delta x = \frac{W - w}{2},$$

yielding

$$\tilde{X}_t^{\text{crop}}(c, u, v) = \tilde{X}_t(c, \ u + \Delta y, \ v + \Delta x), \quad u \in [0, h), \ v \in [0, w).$$

Finally, we resize this crop back to $(H, W)$ using bilinear interpolation $\mathcal{I}_{H,W}(\cdot)$ to obtain the rotated output video $Y \in \mathbb{R}^{C \times T \times H \times W}$:

$$Y(c, t, y, x) = \mathcal{I}_{H,W}\left(\tilde{X}_t^{\text{crop}}(c, \cdot, \cdot)\right)(y, x), \quad y \in [0, H), \ x \in [0, W).$$

## B   MORE IMPLEMENTATION DETAILS

Based on noise initilization strategy introduced in Sec. 3.3, we skip the first $k$ denoising steps that controls the trade-off between style strength and motion fidelity. By trial and error, we choose different $k$ values for each style presented in Table 3. For styles such as Vector that are more abstract, we use less $k$ value and for styles such as Pixar that more resembles the input RGB, we use higher value. For R Precision, we employ Gemini-2.5-Flash as the style classifier.

Table 3: Step skip values used for different styles.

| Style | Step Skip Value |
| --- | --- |
| Vector | 1 |
| 3D Chibi | 2 |
| Anime | 3 |
| Pixar | 6 |
| Clay | 0 |
| LEGO | 2 |
| Macaron | 2 |
| Origami | 2 |
| Rick & Morty | 0 |

## C   STYLE GENERALIZATION

Throughout training we use nine styles to train our style adapter. Table 4 shows quantitative evaluation of PICKSTYLE and other baselines on six new styles: Cyberpunk, Picasso, Wooden Puppet, The Simpsons, Watercolour Pastel, and Minecraft. Still, across all evaluated metrics, PICKSTYLE outperforms prior methods, indicating that it generalizes its style-transfer process effectively beyond the training styles. Fig. 9 qualitatively demonstrates that PICKSTYLE can apply a variety of previously unseen styles to source videos while preserving motion coherence and structural fidelity.

## D   STYLE ALIGNMENT VS. INFERENCE COST TRADE-OFF

Fig. 10 shows that our method achieves both faster inference and better CSD score for style alignment, whereas Rerender and FRESCO rely on Ebsynth blending (Jamriška et al., 2019), which introduces the main bottleneck during inference.

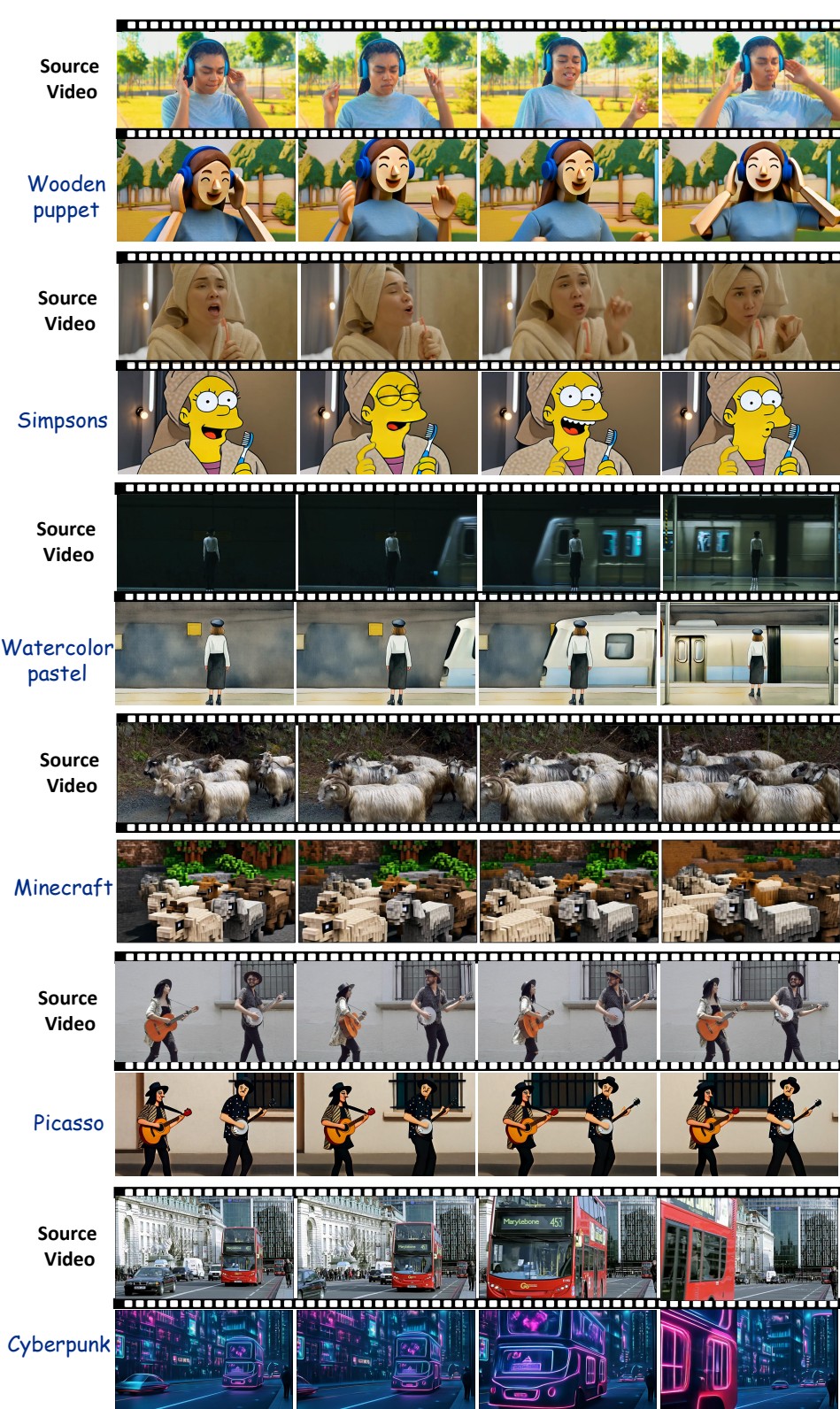

Figure 9: Qualitative results demonstrating PickStyle's generalization to unseen styles.

Table 4: Quantitative comparisons on Content and Style Alignment across baselines and PICKSTYLE on six styles not seen throughout training.

| Models | Content Alignment | | Style Alignment | | | | |
| | DreamSim ↓ | UMT ↑ | CLIP ↑ | CSD ↑ | R Precision ↑ | | |
| | | | | | Top@1 | Top@2 | Top@3 |
|---|---|---|---|---|---|---|---|
| Rerender Yang et al. (2023) | 0.41 | 2.51 | 0.56 | 0.20 | 0.28 | 0.43 | 0.57 |
| FLATTEN Cong et al. (2024) | **0.40** | 2.57 | 0.56 | 0.21 | 0.35 | 0.53 | 0.61 |
| PICKSTYLE | 0.41 | **2.95** | **0.60** | **0.35** | **0.70** | **0.81** | **0.89** |

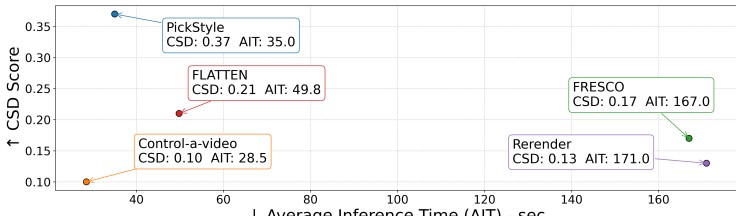

Figure 10: Comparison on CSD Score and inference cost, per one second of generated video. Inference is evaluated on a single H100 GPU.

## E MORE COMPARISON WITH VACE

Alternative conditions that VACE can use for style transfer include depth maps, shown in Fig. 11, and scribbles, shown in Fig. 12. However, because depth maps only provide relative geometry and scribbles capture edges, VACE is unable to perform effective style transfer in either case. Moreover, since these conditions are extracted from videos, they are prone to noise, which further degrades the quality of the generated output.

**Why VACE cannot perform style transfer.** In the original VACE architecture, conditioning inputs are marked as either *reactive* or *inactive*. Reactive modalities (e.g., depth, edges, segmentation, pose) can influence the internal DiT layers, whereas inactive modalities only participate in shallow cross-attention and cannot modify the denoising trajectory. Importantly, RGB video frames are treated as *inactive* conditioning, meaning they do not enter the self-attention pathway and therefore cannot guide geometry-changing style transformations.

We insert lightweight LoRA-based style adapters into the self-attention layers of the VACE context blocks, converting RGB conditioning into a reactive modality. This allows the input video to influence the internal feature evolution of the diffusion model, enabling geometry-aware style transfer and temporal consistency that the original VACE backbone cannot achieve.

## F MORE QUALITATIVE COMPARISON

Additional qualitative comparisons are shown in Fig. 17 and Fig. 18, covering Pixar, 3D Chibi, Origami, Vector, Clay, Macaron, and Rick & Morty styles. Across these diverse cases, competing approaches frequently suffer from color artifacts, style distortion, and unstable temporal consistency. For instance, methods like Rerender and FRESCO often introduce flickering due to their image-based design, while Control-A-Video and FLATTEN struggle to maintain coherent color reproduction and consistent geometry when translating styles across frames. In contrast, PICKSTYLE produces results that remain faithful to the source video while accurately reflecting the intended target style, demonstrating stronger robustness across both simple and complex stylizations.

## G ROBUSTNESS AGAINST CHANGES IN 3D

Fig. 13 illustrates the robustness of PICKSTYLE to 3D viewpoint changes during style transfer. Although the training pipeline only applies 2D warping operations such as zooming and sliding to

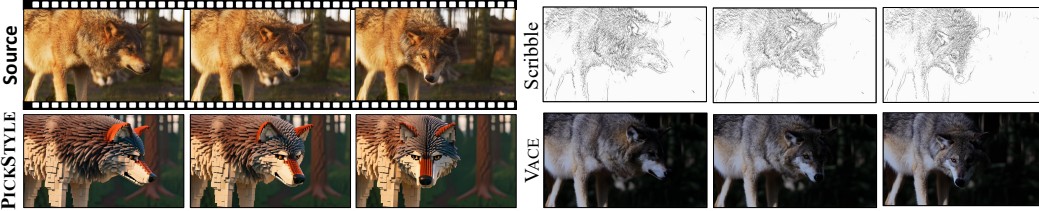

Figure 11: Comparison between PICKSTYLE and the VACE baseline in Anime style when using Depth map as condition of VACE.

Figure 12: Comparison between PICKSTYLE and the VACE baseline in LEGO style when using scribble as condition of VACE.

preserve motion priors, the model still handles genuine 3D variations including large yaw rotations that cannot be simulated by any 2D warp. This shows that our model works reliably even under significant 3D perspective changes.

## H MORE ABLATION ON CS–CFG

The CS–CFG proposed in Section 3.4 relies on the two hyperparameters $t_{\text{guide}}$ and $c_{\text{guide}}$. As shown in Fig. 14, setting $c_{\text{guide}} = 1$ (i.e., not applying the guidance term) produces a faded and weakly defined rod, as highlighted in the zoom-in region. Increasing the guidance strength makes the rod progressively clearer and more faithful to the source structure. However, using overly large guidance values leads to oversaturation in the generated stylized video.

Fig. 15 illustrates the effect of $t_{\text{guide}}$ on the generated video. When the guidance is not applied (i.e., $t_{\text{guide}} = 1$), the output appears less stylized and lacks the intended anime look. Increasing $t_{\text{guide}}$ progressively strengthens the style, while excessively large values again lead to oversaturation in the final generation.

## I ABLATION ON NOISE INITIALIZATION STUDY

Fig. 16 illustrates how skipping the first $k$ denoising steps affects the quality of the generated video. When no steps are skipped and sampling begins from pure noise, the model may produce motions that are not perfectly aligned with the conditioning video. As more initial steps are skipped and the process instead starts from a noisy version of the RGB context, alignment with the original motion improves. For example, with $k = 2$, the workout action is better preserved, and with $k = 4$, the model more accurately retains the people on the treadmill. However, skipping too many steps (e.g., $k = 4$) can negatively impact the applied style, especially when the target style differs significantly from the appearance of the original RGB video.

## J COMPOSITE-STYLE PROMPT

Fig. 19 illustrates how using multiple text-prompt styles affects the output. When several styles are provided, the generated video tends to reflect a weighted blend of them, with the first style having the strongest influence.

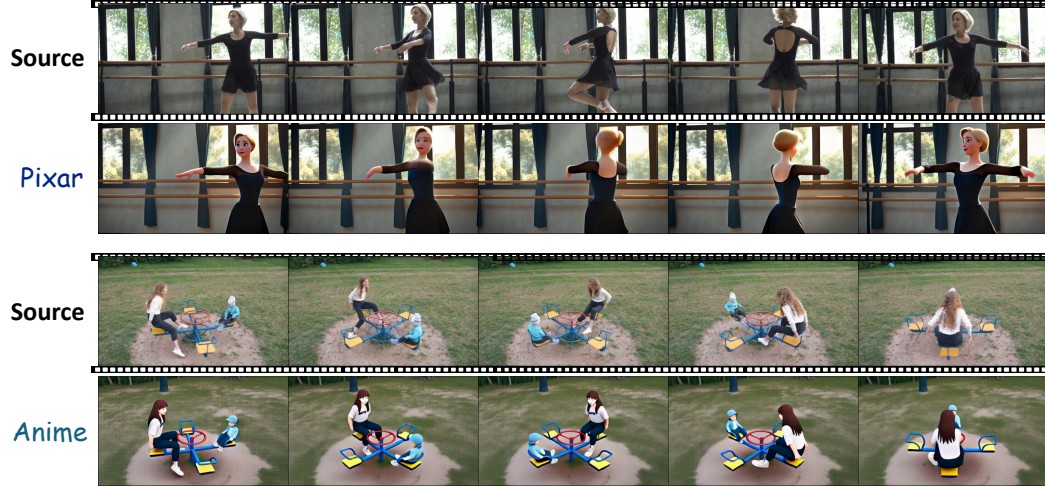

Figure 13: Evaluation samples for 3D-aware consistency. Each sequence shows a subject or object undergoing 3D rotation or viewpoint change, and the stylized outputs (Pixar and Anime) are used to assess whether the method preserves stable 3D structure, consistent geometry, and coherent appearance across frames while changing only the visual style.

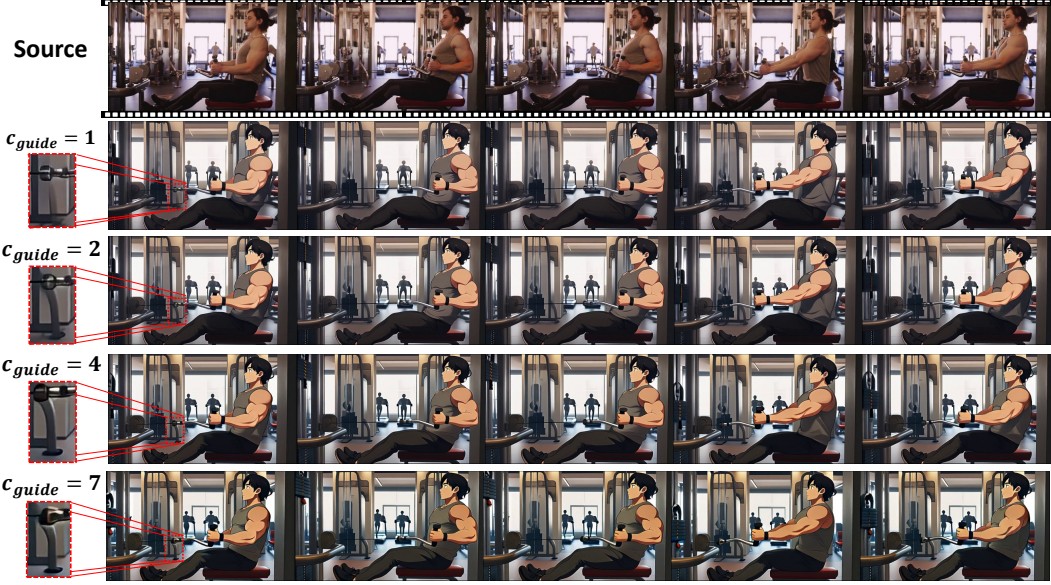

Figure 14: Effect of $c_{\text{guide}}$ on the generated video. $t_{\text{guide}} = 5$ for all generations.

## K  EXTENDED RELATED WORKS

**Video style transfer with image prior.** There are several models that leverage image-based diffusion models for video style transfer by extending them with temporal mechanisms. ControlVideo (Zhang et al., 2023b) adapts ControlNet from images to videos by adding full cross-frame self-attention and interleaved-frame smoothing, which allows strong structural fidelity under text-and-condition guidance. However, it is heavily reliant on the quality of control signals (such as depth or edges), making it less robust when such guidance is noisy or unavailable. ReRender-A-Video (Yang et al., 2023) generates stylized key frames with hierarchical cross-frame constraints using an image diffusion model, and then propagates them to the full video through patch-based blending. This hybrid design balances efficiency and quality but can introduce blurred details or artifacts when large motion or scene changes occur. FRESCO (Yang et al., 2024) builds on im-

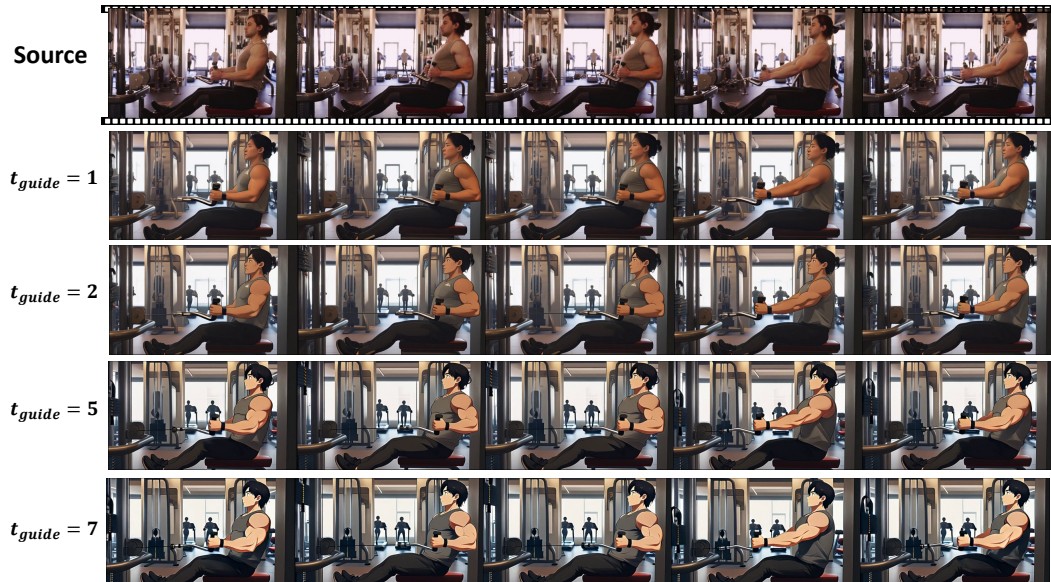

Figure 15: Effect of $t_{\text{guide}}$ on the generated video. $c_{\text{guide}} = 4$ for all generations.

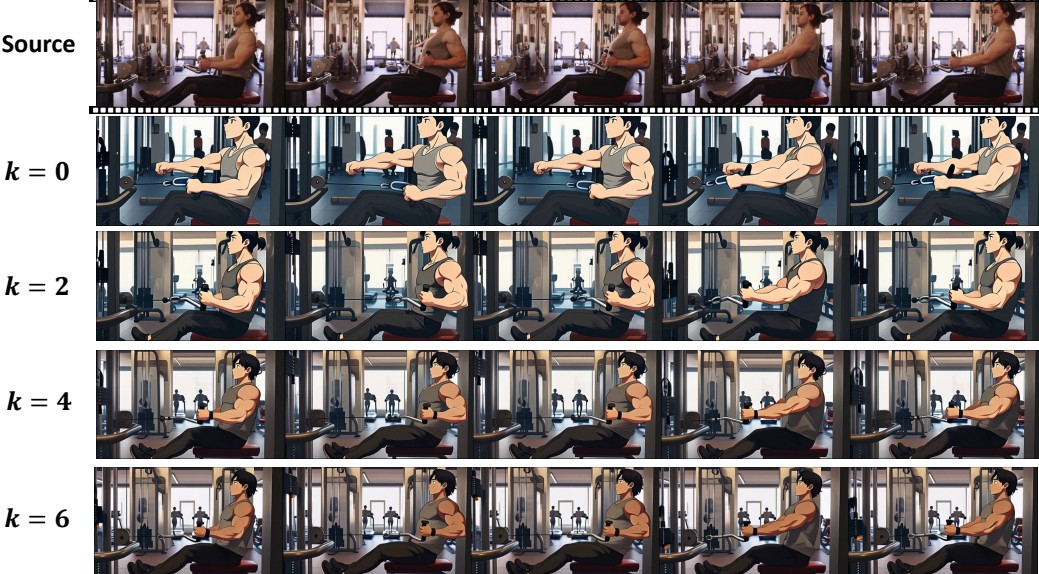

Figure 16: Effect of skipping $k$ initial noising steps in noise initialization strategy.

age priors by enforcing spatial and temporal correspondences and introducing a feature blending mechanism that aggregates spatially similar regions and propagates them along optical flow paths. While this reduces flicker and improves motion stability, it remains sensitive to flow errors and adds computational complexity. FLATTEN (Cong et al., 2024) also introduces flow-guided attention to improve temporal consistency, but still depends on optical flow and degrades when large geometric deviations occur between source and target styles. Despite their progress, all these image-based approaches still find it challenging to fully preserve the natural motion of the input video without noticeable flicker.

**Diffusion-based video editing.** Another related direction involves diffusion-based video editing, where models improve temporal stability by manipulating latent features or deformation fields. Tune-A-Video (Wu et al., 2023) enforces consistency by fine-tuning an image diffusion model on a

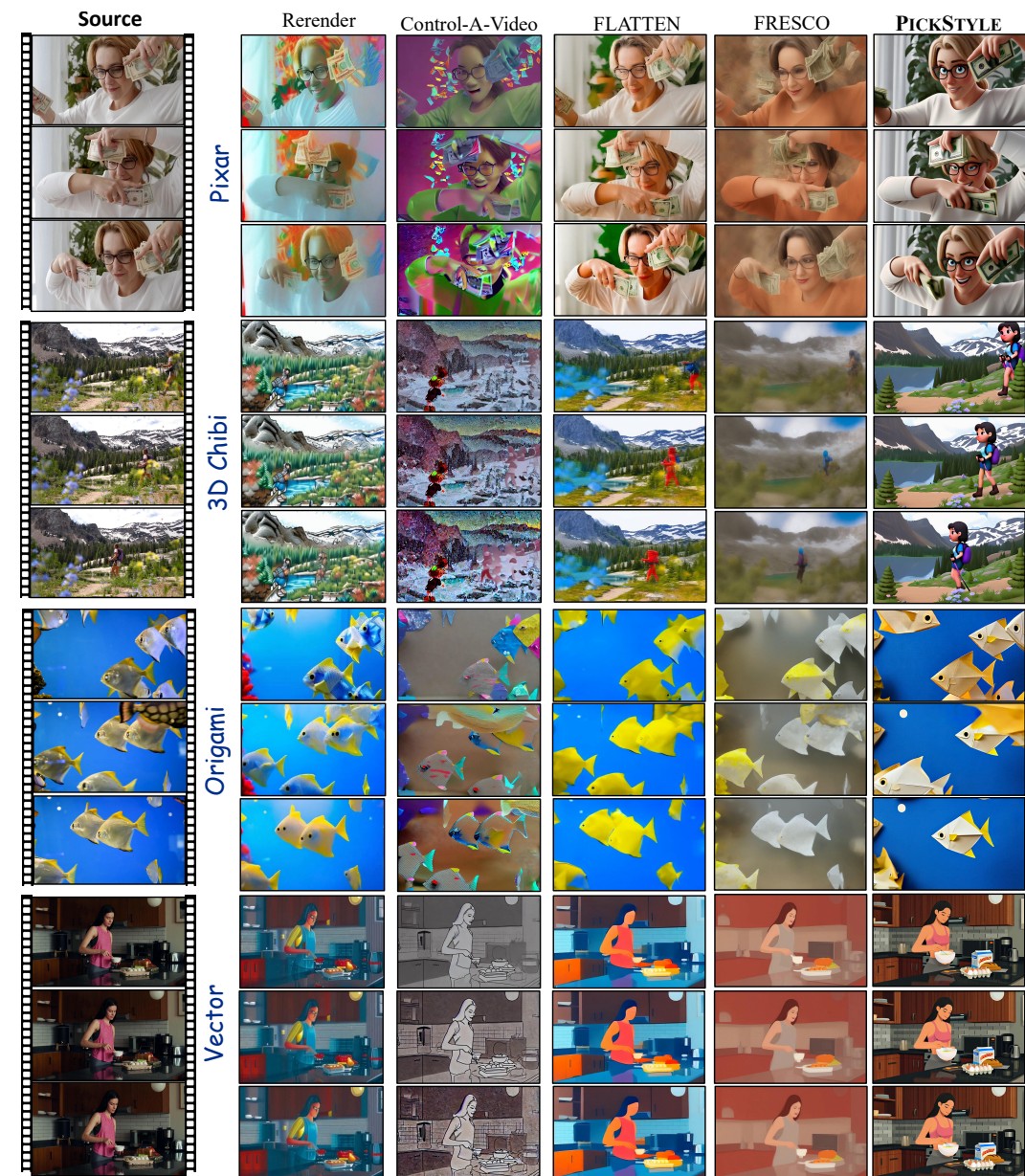

Figure 17: Qualitative comparison in Pixar, 3D Chibi, Origami, and Vector styles.

single video. FateZero (Qi et al., 2023) transfers attention maps across frames to maintain structural alignment during edits. CoDeF (Ouyang et al., 2024) learns deformation fields that warp a canonical representation for coherent frame synthesis. However, these methods are designed for temporally consistent text-guided edits rather than full video-to-video style transfer. Because they rely on the original scene geometry, they struggle with styles that reshape objects or require structural changes across the entire frame. VideoP2P (Liu et al., 2024) adapts prompt-to-prompt editing to video by matching cross-frame attention maps. Align Your Latents (Blattmann et al., 2023) enforces temporal coherence by synchronizing latent trajectories in the diffusion process. VidEdit (Couairon et al., 2023) achieves consistent edits by optimizing a shared latent representation that governs all frames. These methods still rely on standard classifier-free guidance, which mixes semantic and structural signals and limits precise style control

**Video style transfer with video diffusion models.** Only a few works adapt video diffusion models directly for style transfer. Control-A-Video (Chen et al., 2023) extends an image diffusion backbone

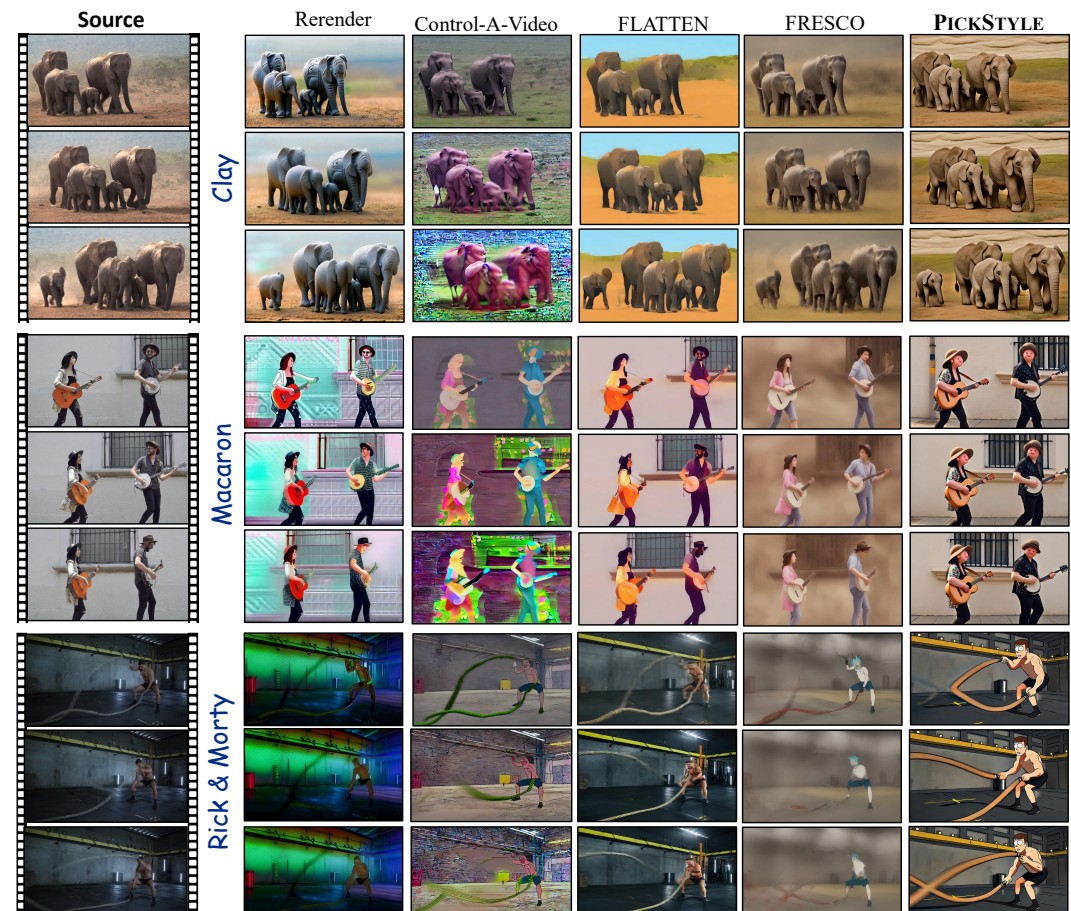

Figure 18: Qualitative comparison in Clay, Macaron, and Rick & Morty styles.

with temporal layers and spatio-temporal attention, and incorporates motion-aware initialization and first-frame conditioning while also supporting per-frame controls such as edges, depth, or flow maps; this allows it to preserve structure and motion while applying styles described in the prompt, though its outputs are generally constrained to short clips and moderate resolutions. V-Stylist (Yue et al., 2025) approaches the problem as a multi-agent pipeline: it parses the input video into shots, interprets an open-ended style request with an LLM, and renders each shot with a style-specific diffusion model and multiple ControlNets, guided by a self-refinement loop that balances style and structure. This design makes it effective for long and complex videos while producing strong style fidelity. StyleMaster (Ye et al., 2025), in contrast, integrates both local and global style cues into a video diffusion backbone, employs a motion adapter to enhance temporal consistency, and uses a tiled ControlNet for video-to-video translation; its styles are often more artistic, as they are grounded in a curated training dataset created using VisualAnagrams, which emphasizes distinctive painterly and creative effects.

Beyond these dedicated stylization models, large video diffusion backbones such as Hunyuan-Video (Kong et al., 2024), LTX-Video (HaCohen et al., 2024), and Wan (Wan et al., 2025) demonstrate high-quality video generation and strong temporal coherence, but they are trained primarily for text-to-video generation rather than video-conditioned style transfer.

**Positioning relative to prior work.** PICKSTYLE differs from image-prior methods by removing reliance on depth, edges, or optical flow, which allows it to handle styles that introduce strong geometric changes. Compared to video diffusion approaches, our framework requires only a single pretrained backbone and introduces lightweight style adapters inside the conditioning branch rather than retraining or managing multiple style-specific models. Our Context–Style Classifier-

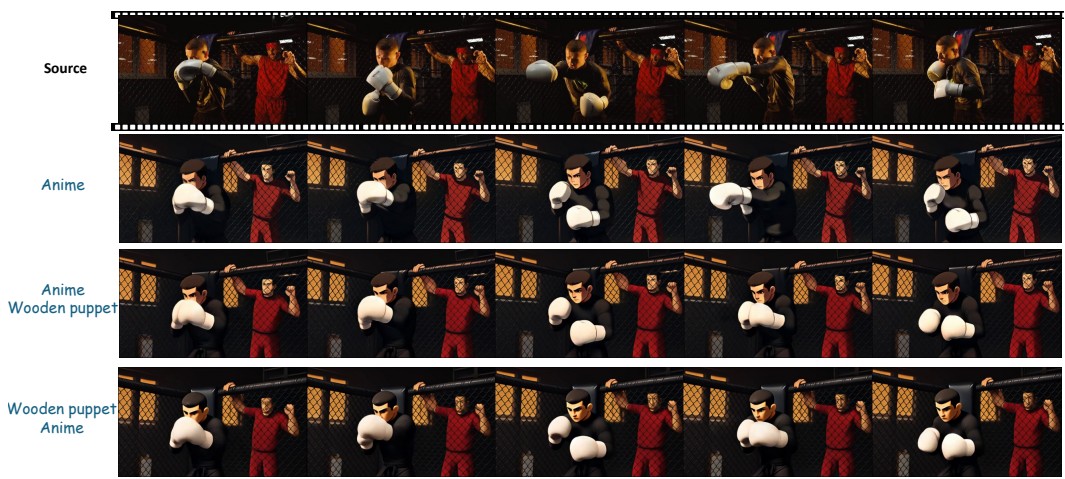

Figure 19: Effect of using multiple styles for text prompt. When multiple styles are applied to a text prompt, the first style tends to dominate the overall effect.

Free Guidance further separates text-driven style and video-driven content directions, addressing the entanglement present in standard CFG-based editing methods.

## L    LIMITATION

PICKSTYLE is built on Wan2.1 as the underlying generative backbone and therefore inherits artifacts and weaknesses present in that model. Typical issues include distortions in fine regions such as faces and hands, where the base model struggles to capture small details. As more advanced video backbones become available, the same pipeline can directly benefit from them, reducing such artifacts and further improving overall quality.

## THE USE OF LARGE LANGUAGE MODELS (LLMS)

We use GPT-5 to refine the writing, paraphrase content, and improve readability.

