# OpenReview forum: "PickStyle: Video-to-Video Style Transfer with Context-Style Adapters"
_ICLR.cc/2026/Conference — Submitted to ICLR 2026_

### Official Review · Reviewer_zJFo · 2025-10-28

**Soundness:** 3
**Presentation:** 3
**Contribution:** 2
**Rating:** 6
**Confidence:** 4

**Summary:**

This paper introduces PICKSTYLE, a video style transfer framework that efficiently adapts pre-trained video diffusion models via low-rank style adapters and a novel guidance strategy, achieving superior results by leveraging paired image data and simulating temporal coherence.

**Strengths:**

The Core Innovations are as follows:

1. Efficient Low-Rank Adaptation: The integration of specialized style adapters into self-attention layers enables effective motion-style transfer while maintaining computational efficiency and strong content-style alignment.

2. Bridging Image-Video Domain Gap: A novel synthetic clip construction strategy using shared augmentations that simulate camera motion, effectively leveraging static image supervision for dynamic video stylization.

3. Factorized Classifier-Free Guidance: The proposed Context-Style Classifier-Free Guidance (CS-CFG) innovatively disentangles style and context control, ensuring precise style application without compromising video content integrity.

**Weaknesses:**

1. The proposed 2D motion simulation may lack generalizability for complex real-world videos involving significant 3D perspective changes. The current experiments, as shown in Figure 7, are primarily validated on relatively simple motions, leaving its performance on more complex scenarios unverified.

2. The decision to keep the text-video cross-attention module frozen is a potential limitation. If the base model was not exposed to certain style descriptions during its pre-training, the framework might struggle to establish a correct correspondence between novel textual style prompts and their visual manifestations.

**Questions:**

Could you clarify the design choice for C_null? Specifically, why was shuffling frame orders adopted instead of using a null context (empty set)? Furthermore, what is the rationale behind the specific form of the context direction, why don't use this formulation as the context direction: ϵ_cond - ϵ_theta(x_t, t; T, C_null), and were other alternative formulations explored or ablated?

---

> ### Author Response · Authors · 2025-11-22
>
> # W1
> **Evaluation Under 3D Motion and Perspective Changes:** We included Section G in the supplementary material, where we evaluate two videos exhibiting 3D changes: a ballet dancer rotating in the first video and a merry-go-round in the second. In both cases, PICKSTYLE demonstrates consistent style transfer performance.
>
> # W2
> **Rationale for freezing Text–Video attention:** We emphasize that our goal in PICKSTYLE is not to teach the base model entirely new styles, but to teach it the process of style transfer in a video-to-video setting while staying coherent with the input RGB video. For this reason, we keep the text–video cross-attention module frozen.  This preserves the strong style priors learned by Wan during large-scale training and ensures that PICKSTYLE focuses on learning (i) how to map an input video into the style manifold of the base model, and (ii) how to maintain coherence with the RGB conditioning video in both motion and visual appearance (e.g., object identity, clothing colors, scene layout, etc.). Therefore, PICKSTYLE is able to generalize to other styles (see section C of supplementary) as it retains the base model's ability to generate styles beyond those used during our specific fine-tuning, as its language-to-visual mapping remains intact. While not the primary aim of this paper, the framework could also be extended to inject new style priors.
>
> # Q1
> **Design choice for C_{\text{null}}​ and context direction in CS–CFG:** Figure 7 shows that when shuffling is not used, combining context and text guidance leads to stronger oversaturation in the generated results. In addition, shuffling can sometimes improve style transfer; for example, the person’s hand in the clay style output of Figure 7 appears noticeably better.
>
> As for the choice of context and text directions in CS--CFG, other formulations could also work. However, we empirically find that our proposed design produces higher-quality results. Intuitively, preserving context during generation is more challenging because the model can remove objects or small details to better align with the conditioning input. With our proposed context direction $\epsilon_\theta(x_t, t;\, \varnothing, C) - \epsilon_\theta(x_t, t;\, \varnothing, C_{\text{null}}$)
> the difference between the conditioned and unconditioned predicted noise is generally larger since no text prompt is provided. In contrast, the alternative $\epsilon_{\text{cond}} - \epsilon_\theta(x_t, t;\, T, C_{\text{null}}$)
> captures differences only when the text prompt is present, making the model less flexible to deviate compared to the unconditional case. For the text guidance term, we include the context $C$ in both predictions because our goal is to emphasize only the style effect while keeping the context fixed.

---

### Official Review · Reviewer_PThQ · 2025-10-31

**Soundness:** 2
**Presentation:** 2
**Contribution:** 2
**Rating:** 4
**Confidence:** 4

**Summary:**

The paper introduces PickStyle, a video style transfer framework using diffusion models to render videos in a text-specified style while preserving content, overcoming the lack of paired video data. It achieves this by augmenting video diffusion backbones with low-rank style adapters and training on synthetic video clips created from paired images using simulated camera motion. Context–Style Classifier-Free Guidance (CS–CFG) is also introduced, which independently guides the style (text) and context (video) directions, resulting in superior, temporally coherent, and content-preserving style translations.

**Strengths:**

The paper presents end-to-end, feedforward video style transfer network.
The dataset construction strategy would be useful to the research community.

**Weaknesses:**

- All baseline methods are built on far inferior backbones (some are based on t2i backbone) and it’s not a surprise that PickStyle-based on VACE (WAN) beats the selected baselines. More recent baselines or any methods thats applied on the same WAN backbone is needed.
- The base model, VACE inherently cannot perform video style transfer? If so, can we see how PickStyle is improved compared to the original VACE backbone?
- Compared to the normal CFG, how much more computation overhead does CS-CFG incur in terms of both memory and time?
- My biggest concern is the lack of technical contribution. The adapter module is ControlNet-style network, also frequently adapted and used in recent DiT-based generation methods. The CFG with additional condition term (triangular CFG) is not new either. For example, SV3D or VideoJAM uses similar approaches.

**Questions:**

Please see the weakness above

---

> ### Author Response · Authors · 2025-11-22
>
> # W1
> **Rationale for baseline choice:** The baselines we evaluate reflect the complete set of publicly available methods that can realistically perform video-to-video style transfer. Existing approaches fall into two categories: (i) image-prior diffusion methods which we chose methods that represent the strongest models currently used for stylization with published code to test on the same video benchmark, and (ii) diffusion-based video models that can accept conditioning videos. Control-A-Video and VACE are the only video diffusion backbone that supports video conditioning, but they do not take RGB as condition and VACE also does not contain any module that enables style transfer; PICKSTYLE is the first to make this capability possible.
> Importantly, there are no existing video-to-video stylization methods built on WAN or VACE, and no prior work is capable of geometry-level style transfer (not texture only) at the video level. Our comparison therefore includes the strongest available image-prior baselines and the only feasible video-conditioned diffusion backbone.
>
> # W2
> **Comparison Between PICKSTYLE and VACE Backbone:** VACE inherently cannot perform video style transfer with RGB frames as reactive input. It is limited to text-guided editing or using non-RGB conditions like optical flow or depth maps. We have included comparisons against VACE using its compatible modalities (Optical Flow, Depth, Scribble) in Figs. 6, 11, and 12, where it fails to capture the target style. This shows that PICKSTYLE's style adapter is necessary to enable the task on the state-of-the-art backbone for this task.
>
> # W3
> **CS–CFG Overhead:** The computational overhead for CS-CFG is minor: it adds one extra forward pass (ϵ_null_text) compared to standard conditional CFG, which requires two passes (ϵ_cond and ϵ_null), resulting in a total of three forward passes. This marginal increase in inference time is justified by the significant and measurable gains in style fidelity and dynamic quality (Table 2).
>
> # W4 (Technical Contribution)
>
> **Role of style adapter:** Our adapters differ fundamentally in both placement and function. ControlNet-style network is an external auxiliary branch that predicts residual features (e.g., depth, edges, pose) and injects them into a frozen diffusion model. It is designed in the baseline of our backbone (VACE) is not a contribution of our work. Moreover, It is not designed to learn appearance-to-appearance mappings or geometry-altering transformations. In contrast, our style adapters are inserted directly into the spatiotemporal self-attention layers of the context branch that process the RGB video-conditioning pathway.
>
> **Training on still images:** We mitigate the challenge of lacking paired stylized video data by introducing a solution that enables training on still images for moving-video stylization. Unlike previous approaches that have separate spatial and temporal attention blocks and decompose the training, we use backbone that has spatiotemporal attention that makes the previous approaches obsolete. Instead, we generate dynamic clips through synthetic camera motions, allowing the model to retain temporal priors and generalize from static image supervision to dynamic video content.
>
> **Factorized CS-CFG:** Our CS-CFG is designed to disentangle the conflicting demands of style (text) and content/coherence (video). Unlike a simple "triangular CFG," the key innovation is the construction of $\Delta_{context}$ by using a _spatiotemporal permutation_ of the video features ($\mathcal{C}_{null}$ defined explicitly in Eq. 10). This allows the context direction to specifically push the sample towards the video's motion and structure, independent of the style signal. The ablation (Fig. 8) proves this specific design is essential for achieving superior content and temporal fidelity compared to standard CFG or a naive "zero context" null. Neither SV3D nor VideoJAM do this.
>
> We revised the paper and clarified our contributions in the last paragraph of the introduction.

---

### Official Review · Reviewer_eovg · 2025-11-01

**Soundness:** 3
**Presentation:** 3
**Contribution:** 3
**Rating:** 2
**Confidence:** 4

**Summary:**

PickStyle introduces a video-to-video style transfer framework that preserves motion and context while rendering stylized frames from one of nine trained styles. The method’s primary innovation seems to be a context-style classifier-free guidance mechanism, allowing explicit control over content and style conditioning during diffusion. Additionally, a tunable noise initialization strategy enables improved temporal coherence and perceptual fidelity. The paper includes reasonable experiments demonstrate improvements over prior methods in both qualitative and quantitative metrics, including a standard battery of metrics across content, video quality, etc.

**Strengths:**

S1 The paper builds on the Wan2.1 generation backbone to include style adaptation and seems to be able capture nine different styles in a way that generalizes from image-pairs to video.  The paper makes reasonable technical innovations to accomplish this that seem original and relevant, such as CS-CFG which creates a tunable trade-off between fidelty and stylization (although this trade-off seems not analyzed in the paper).

S2 The paper includes a motion augmentation strategy that enables the use of image pairing as training data.

S3 Results seem compelling and meaningful analyses are included.  It seems clear that PickStyle is best at being able to match the style prompt, at least according R Precision score (although it is curious that this score only uses one frame from the video).  It also seems that the video quality aspects are strong.

**Weaknesses:**

W1 The key aspects of the proposed method seem to the LoRA adapters that modulate the attention to capture style, the CS-CFG method and the noise initialization.  Yet, none of these are really thoroughly analyzed in any way.  For example although Fig. 8 captures one example of where CS-CFG helps, the interplay between $t_\text{guide}$ and $c_\text{guide}$ is not studied.  Similarly, we have no evidence about to what degree the context-based initialized is necessary.  Hence it is impossible to actually assess whether the technical innovations align with the observed results improvements, or if it is from other sources (e.g., the different data, the augmentation approach, etc.)

W2 It is not clear whether the comparisons are fair.  Considering this paper creates a composite dataset with nine styles, have the other methods to which the paper compares been retrained on this dataset?  The paper does not sufficiently describe this critical point.

W3 The approach to augment the paired image samples with some motion to generating pair training videos is not well described in the text and therefor hard to analyze.  It would seem, for example, that the types of augmentations used are not able to capture realistic motions in video resulting from 3D content and perspective effects.  This implies that perhaps the datasets used and results shown, however compelling they may be, may not be indicative of utility on more general video.

W4 It seems that PickStyle is the most computationally expensive of the methods evaluated.


Minor things
- The manner in which the references are typically cited, e.g., "VACE Jiang et al. (2025)" is not proper, at least not for this style of including the author name.  These should be in parenthesis or better incorporated directly into the text.  VACE by Jiang et al. (2025) or VACE (Jiang et al. 2025).

**Questions:**

Q1 What would happen if multiple style prompts were given as input?  What would happen if an out of set style prompt were given?

Q2 What are the limitations of applying this to video?  Is there any reason to expect degradation for longer videos, for example?

Q3 Is the dataset created here publicly available?

---

> ### Author Response · Authors · 2025-11-22
>
> # W1
> **Ablations on CS–CFG and Noise Initialization**
>
> We added Section H in the supplementary material to qualitatively evaluate the effect of varying $t_{\text{guide}}$ and $c_{\text{guide}}$ on the generated video (shown in Figures 14 and 15). In addition, we included Section I in the supplementary material to present an ablation study on the noise initialization strategy.
> Together, these analyses show that both CS–CFG and the context-based initialization contribute meaningfully and independently to temporal coherence and style–content balance, confirming that the observed improvements stem from the proposed components rather than data or augmentation alone.
>
> # W2
> **Fair Comparison and Role of Training Styles**
>
> Thank you for the constructive feedback. In Section C of the supplementary material, we evaluate PICKSTYLE using six new styles that were not seen during training. We compare our method quantitatively against the two best-performing baselines from the main paper and qualitatively demonstrate that PICKSTYLE generalizes well to unseen styles. We also highlight that we use Wan2.1 as a strong video model with robust style priors.
> Our contribution focuses on style-conditioned video generation without sacrificing the model’s motion prior, even though the adapter is trained on static images. We also want to emphasize that our goal in PICKSTYLE is not to teach the base model entirely new styles, but to teach it the process of style transfer in a video-to-video setting while staying coherent with the input RGB video. So the styles used during training serve only as exemplars to teach the model the underlying style-transfer process.
> Our method preserves the strong style priors learned by Wan during large-scale training and ensures that PICKSTYLE focuses on learning (i) how to map an input video into the style manifold of the base model, and (ii) how to maintain coherence with the RGB conditioning video in both motion and visual appearance (e.g., object identity, clothing colors, scene layout, etc.). Therefore, PICKSTYLE is able to generalize to other styles as it retains the base model's ability to generate styles beyond those used during our specific fine-tuning.
>
> # W3
>
> **Elaboration on the augmentation strategies** . We added Section A in the revised manuscript to provide a detailed explanation of the applied augmentations.
>
> **Evaluation under 3D motion and perspective changes.** We included Section G in the supplementary material, where we evaluate two videos exhibiting 3D changes: a ballet dancer rotating in the first video and a merry-go-round in the second. In both cases, PICKSTYLE demonstrates consistent style transfer performance.
>
> # W4
>
> We agree that Wan2.1 requires significantly more computation compared to other baselines. However, as shown in Figure 9, the average inference time for generating one second of video with PICKSTYLE remains lower than that of the other methods. This is primarily because prior works rely on Ebsynth blending to merge generated frames into a video, which adds substantial overhead.
>
>
> # Q1
> We added Section K in the supplementary material to evaluate this scenario. We tested prompts written as “anime style. wooden puppet style.” and “wooden puppet style. anime style.” and compared them with the baseline prompt “anime style.” In both cases, the generated video converges to a blend of the two styles, but the first style listed consistently has the stronger influence.
>
> # Q2
> Longer video generation is not possible due to limitations of Wan2.1. However, recent works such as Self-Forcing++ [1] extend the generation of Wan2.1 into longer videos. Same approaches can also be applied to PICKSTYLE for longer style transfer.
>
> [1] Cui, J., Wu, J., Li, M., Yang, T., Li, X., Wang, R., Bai, A., Ban, Y. and Hsieh, C.J., 2025. Self-Forcing++: Towards Minute-Scale High-Quality Video Generation. arXiv preprint arXiv:2510.02283.
>
> # Q3
> Yes. We release the model weights, dataset, training and demo code upon acceptance.

---

### Official Review · Reviewer_tRTK · 2025-11-01

**Soundness:** 2
**Presentation:** 1
**Contribution:** 2
**Rating:** 2
**Confidence:** 3

**Summary:**

This paper introduces PICKSTYLE, a video-to-video style transfer framework leveraging diffusion models with context-style adapters. The method aims to preserve motion and context while translating videos into diverse styles, using paired still image data and synthetic motion augmentation for training. A novel Context–Style Classifier-Free Guidance (CS–CFG) mechanism is proposed to independently control style and context during generation. The approach is evaluated against several baselines, showing improvements in temporal coherence, style fidelity, and perceptual quality across multiple metrics and styles.

**Strengths:**

1) The qualitative and quantitative results demonstrate that PICKSTYLE achieves superior style transfer, temporal stability, and perceptual quality compared to existing baselines.
2) The paper provides extensive quantitative and qualitative comparisons with multiple baselines, covering a wide range of styles and metrics.

**Weaknesses:**

1) The paper is difficult to follow in several key sections. The training procedure, especially how style and content consistency are achieved, is not clearly explained. The technical details of the model architecture and training pipeline are scattered and could benefit from a more structured presentation.
2) The manuscript does not sufficiently highlight the core technical differences that make PICKSTYLE outperform baseline methods. The related work section is shallow, mostly listing existing approaches without deep analysis or positioning of the proposed method’s unique contributions.
3) The training dataset is selectively curated, focusing on a limited set of styles (e.g., Anime, Pixar, Clay, LEGO, etc.) and synthetic Unity3D renderings. There is little discussion or evidence regarding the model’s ability to generalize to styles not covered in the training data, raising concerns about robustness and applicability.
4) It is unclear whether baseline methods were trained or fine-tuned on the same dataset as PICKSTYLE. Without this information, the fairness of the comparisons and the claimed superiority of the proposed method are questionable.

**Questions:**

Refer to the weakness part

---

> ### Author Response · Authors · 2025-11-22
>
> # W1
> **Clarification and reorganization of the training pipeline**
>
> We substantially revised Section 3 by restructuring it into a coherent sequence (preliminaries → style adapter → synthetic training clips → CS–CFG → training summary), adding a high-level summary paragraph (see Sec. 3.6), and tightening the explanation of how style and content consistency are achieved through our RGB-conditioned style adapters, synthetic motion-augmented training clips, and the factorized CS–CFG formulation.
> Moreover, Figure 2 provides a complete overview of the architecture and training pipeline, showing where each component introduced in Section 3 is applied.
>
> # W2
> **Highlighting core contributions and relation to prior methods**
>
> In the revised manuscript, we emphasized in the first paragraph of Section 3.2 why previous methods struggle with complex style transfer. Their main limitation is that they depend on depth or edge maps as the conditioning modality. In contrast, our proposed style adapter is the first to successfully use RGB images directly as the conditioning input for video style transfer. We also updated the related work section: Section 2 now includes a brief overview, and a more detailed discussion appears in Section K of the supplementary material. At the end of the extended related work, we added a clearer explanation of how PICKSTYLE fits within and differs from prior research.
>
> # W3 & W4
> **Generalization to Unseen Styles and Fairness of Baselines**
>
> Thank you for the constructive feedback. In Section C of the supplementary material, we evaluate PICKSTYLE using six new styles that were not seen during training. We compare our method quantitatively against the two best-performing baselines from the main paper and qualitatively demonstrate that PICKSTYLE generalizes well to unseen styles. We also highlight that we use Wan2.1 as a strong video model with robust style priors.
> Our contribution focuses on style-conditioned video generation without sacrificing the model’s motion prior, even though the adapter is trained on static images. We also want to emphasize that our goal in PICKSTYLE is not to teach the base model entirely new styles, but to teach it the process of style transfer in a video-to-video setting while staying coherent with the input RGB video. Our method preserves the strong style priors learned by Wan during large-scale training and ensures that PICKSTYLE focuses on learning (i) how to map an input video into the style manifold of the base model, and (ii) how to maintain coherence with the RGB conditioning video in both motion and visual appearance (e.g., object identity, clothing colors, scene layout, etc.). Therefore, PICKSTYLE is able to generalize to other styles (see section C of supplementary) as it retains the base model's ability to generate styles beyond those used during our specific fine-tuning

---

### Author Response · Authors · 2025-11-22

First of all, we would like to express our gratitude to all reviewers for their insightful comments and constructive feedback on our paper PickStyle. We are glad that the strengths of our work are recognized as strong results (tRTK), style adaptation and motion augmentation (eovg), useful dataset construction (PThQ), and efficient adapters and factorized guidance (zJFo). All concerns raised by the reviewers have been addressed, and we provide the revised version with all changes highlighted **in blue** for clarity.

---

### Author Response · Authors · 2025-11-26

Dear ACs and Reviewers,

As we approach the end of the discussion period, we would appreciate any updates from the reviewers and are happy to continue the discussion as needed.

---

### Author Response · Authors · 2025-12-02
**Discussion Summary**

Dear AC,
We have carefully addressed each comment by the reviewers and revised the paper accordingly (highlighted in blue). Here we summarize concerns and how we addressed them. Hopefully this would reduce the burden on AC and helps making the final decision easier.

**Reviewer tRTK**
- **Main Concerns:** Paper hard to follow; unclear training pipeline; unclear novelty vs baselines; unclear generalization; unclear fairness of comparisons.
- **How addressed:** Rewrote Section 3 for clarity; expanded related work & clarified core contributions; added unseen-style experiments; clarified baseline fairness and role of Wan2.1 priors.

**Reviewer eovg**
- **Main Concerns:** Missing ablations for CS-CFG & noise initialization; unclear augmentation; unclear fairness; concerns about 3D motion and compute cost.
- **How addressed:** Added full ablations (Supp. H, I); detailed augmentation (new Section A); added unseen-style and 3D-motion evaluations; showed PickStyle is faster than Ebsynth-based baselines; answered all questions (multi-style prompts, long videos).

**Reviewer PThQ**
- **Main Concerns:** Baselines too weak; unclear comparison to backbone; CS-CFG overhead not quantified; concerns about novelty.
- **How addressed:** Justified baseline selection; added direct comparisons vs VACE showing it cannot do style transfer; quantified CS-CFG cost (one extra forward pass); clarified novelty of adapter placement, synthetic training clips, and CS-CFG design.

**Reviewer zJFo**
- **Main Concerns:** 2D motion augmentation may fail on real 3D motion; freezing text–video attention may limit unseen styles; questions about CS-CFG design.
- **How addressed:** Added 3D-motion evaluations showing strong performance; explained freezing preserves Wan’s priors and supports generalization; clarified $C_{null}$​ and context-direction rationale, with empirical evidence.

---

### Meta-Review · Area_Chair_E5He · 2025-12-04

**Summary:**

The submission received majorly negative ratings. It introduces a video-to-video style transfer framework built upon pre-trained video diffusion models. The method incorporates low-rank style adapters, synthetic training clips from paired images, and a novel Context-Style Classifier-Free Guidance (CS-CFG) mechanism. While the authors present compelling qualitative results and demonstrate improvements over selected baselines, several weaknesses are identified in the reviewing process, including the limited technical novelty contribution compared with existing works, the unfair and unclear setting in the experimental study (the counterparts are weak or incomparable, and it is unclear whether they were trained on the same data or evaluated under equivalent conditions), the limited generalization and evaluation (unseen styles, complex 3D motion, and longer videos). With the current form, it fails to meet the bar for acceptance at ICLR 2026.

**Reviewer Concerns:**

In the rebuttal, the authors provided additional ablation studies for CS-CFG and noise initialization (in response to Reviewer eovg). Evaluation on unseen styles and 3D motion scenario is added in response to Reviewer tRTK, eovg, zjFo. The rebuttal clarified baseline selection and fairness, explaining why stronger baselines were not included, in response to Reviewer PThQ. The authors elaborated on augmentation strategies and computational overhead, addressing Reviewer eovg and PThQ.

There are still outstanding issues. Despite clarifications, the core technical contributions (style adapters and CS-CFG) are still seen as incremental adaptations of existing techniques. The generalization is not fully established. While new experiments were added, the model’s reliance on Wan2.1’s prior limits its ability to handle truly novel styles or complex real-world motion. The fairness in the experimental setting is still in doubt. The use of Wan2.1 as a backbone gives the proposed method an inherent advantage over older baselines, and the absence of comparisons with similarly strong video diffusion models undermines the claimed superiority.

**Reviewer Scores:**

Reviewer tRTK probably will insist on the original rating of 2, due to his/her doubt on methodological depth.

Reviewer eovg might increase from 2 to 4, but I am not sure.

Reviewer PThQ (4) was most critical of novelty and baseline strength. The rebuttal did not fully solve these concerns. I believe he/she will not support the acceptance of this submission.

Reviewer zjFo is the only one who rated this submission positively (6). But his/her concerns were not fully solved.

---

### Decision · Program_Chairs · 2026-01-26

Reject